



# Entangled Dynamos and Joule Heating in the Earth's Ionosphere

Stephan C. Buchert[1]

[1]Swedish Institute of Space Physics, Uppsala, Sweden

**Correspondence:** Stephan Buchert (scb@irfu.se)

**Abstract.** The Earth's neutral atmosphere is the driver of the well-known Solar quiet (Sq) and other magnetic variations, observed since more than 100 years. Yet the understanding of how the neutral wind can accomplish a dynamo effect has been incomplete. A new viable model is presented, where a dynamo effect is obtained only in case of winds perpendicular to the magnetic field $\boldsymbol{B}$ that spatially vary along $\boldsymbol{B}$. Uniform winds have no effect. We identify

Sq as being driven by wind differences at magnetically conjugate points, and not by a neutral wind per se. The view of two different but entangled dynamos is favoured, with some conceptual analogy to quantum mechanical states. Because of the large preponderance of the neutral gas mass over the ionized component in the Earth's ionosphere the dominant effect of the plasma adjusting to the winds is Joule heating. The amount of global Joule heating power from Sq is estimated, with uncertainties, to be much lower than Joule heating from ionosphere-magnetosphere coupling at

high latitudes in periods of strong geomagnetic activity. However, on average both contributions could be relatively evenly matched. The global contribution of heating by ionizing solar radiation in the same height range should be 2–3 orders of magnitude larger.

## 1 Introduction

The interaction between the ionospheric plasma and neutral wind in the Earth's atmosphere has been described

scholarly in textbooks (e. g., Kelley, 2009) and numerous research articles. Still for a long time the author has felt that his understanding of the subject is incomplete. In this work we describe progress that has been finally made when thinking about the solar quiet (Sq) magnetic variations at mid latitudes. A praiseworthy review of Sq has been published recently by Yamazaki and Maute (2017). Sq is driven by a neutral dynamo. Vasyliūnas (2012) has summarized the fundamental equations for a neutral dynamo and his critical view of the understanding within

the community. The conceptual difficulty of the author's interpretation of the neutral dynamo can be phrased less mathematically as follows: In the frame of the neutral gas the product $\boldsymbol{j} \cdot \boldsymbol{E}^*$, $\boldsymbol{j}$ the electric current and $\boldsymbol{E}^*$ the electric field, is in the steady state zero or positive, because of the well-known Ohm's law for the ionosphere. This indicates that Joule heating takes place (which, however, has not been addressed yet in works specifically on Sq, as far as we are aware of). A common comprehension seems to be that the dynamo effect occurs in the Earth-fixed frame where





$\boldsymbol{j} \cdot (\boldsymbol{E}^* - \boldsymbol{u} \times \boldsymbol{B})$, $\boldsymbol{u}$ the neutral wind, can be negative as required for a dynamo. However, a clear justification for choosing this frame over any other of the infinitely many possible frames seems to be lacking.

We will first present a new viable steady state model of the neutral dynamo in the Earth's ionosphere. As the title suggests, it actually involves (at least) two dynamos. A discussion of various aspects of the new model follows,
with also further references to other works on the subject.

## 2   Preliminaries

A scenario is considered where the lower thermosphere within two latitude circles in each hemisphere is connected by the dipolar geomagnetic field, as sketched in Figure 1. In the northern hemisphere branch an eastward (westerly) zonal wind flows, in the southern hemisphere a westward (easterly) wind. A zero tilt between geodetic directions
(westerly, easterly) and a magnetic field aligned cartesion coordinates is assumed. A ionosphere with a dynamo region exists, as well as magnetized plasma in a plasmasphere (not sketched in Figure 1). The plasma adjusts to the conditions imposed by the neutral winds, but does not interfer otherwise, meaning that neither plasma pressure gradients nor electric fields penetrating from outside etc. do play any role. Zero meridional wind is assumed, and the deviation of the magnetic field $\boldsymbol{B}$ from a vertical inclination in the latitude range of the zonal wind is ignored.
The latitude range is small, such that gradients of $\boldsymbol{u}$, $\boldsymbol{B}$ and the height-integrated Pederen conductance $\Sigma_P$ across the range are neglected. In other words, $\boldsymbol{u}$, $\boldsymbol{B}$ and $\Sigma_P$ are assumed constant across the latitudes of the zonal jets. The scenario is highly simplified compared to any realistic one, in order to achieve a good physical understanding of the situation.

We strive only for a steady state description. The neutral wind in the Earth's ionosphere at mid-latitudes changes
only slowly over time scales of several hours, and the plasma between hemispheres would be able to adjust within seconds, practically instanteneously, with only small amplitudes in the transients. We use the jargon and paradigms of the ionosphere community. Astrophysical dynamos (usually without a neutral gas) are typically rather described in terms of a mechanical MHD approach. Differences between the two approaches have been discussed by Parker (1996), for the high latitudes, and by Vasyliūnas (2012) for specifically the Earth's neutral wind dynamo. Both
authors acknowledged that for the steady state both approaches give equivalent results, and that for highly symmetric cases, such as this one, the "traditional" ionospheric $E$ and $j$ paradigm is efficient and mathematically simpler. The electrodynamics of the ionosphere particularly in the steady state and with a neutral gas is scholarly treated in Kelley (2009).

For completeness we rephrase the most important points relevant for this work: An electric field in the reference
frame of the neutral gas $\boldsymbol{E}^*$ drives in the dynamo region, roughly at altitudes 90 km–350 km, where collisions are significant, Pedersen and Hall currents $\boldsymbol{J}_P$ and $\boldsymbol{J}_H$ according to Ohm's law for the ionosphere:

$$\boldsymbol{J}_P = \Sigma_P \boldsymbol{E}^*, \quad \boldsymbol{J}_H = -\Sigma_H \boldsymbol{E}^* \times \boldsymbol{B}/B. \tag{1}$$



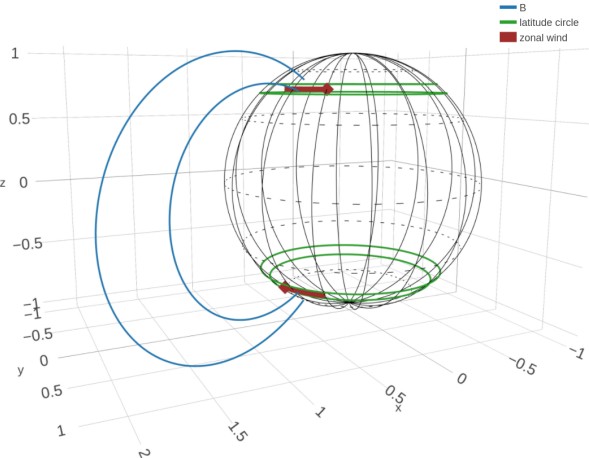

**Figure 1.** 3d sketch of a scenario where regions between $L = 2.5$ and $3$ are magnetically connected by a dipole magnetic field. Supporting information includes this figure in html format which, when opened in a browser supporting JavaScript, allows to change the camera view point.

$\Sigma_P$ and $\Sigma_H$ are the Pedersen and Hall conductances, $\boldsymbol{B}$ the magnetic field. The electric field $\boldsymbol{E}$ in other reference frames with the neutral gas velocity $\boldsymbol{u} \neq 0$ is

$$\boldsymbol{E} = \boldsymbol{E}^* - \boldsymbol{u} \times \boldsymbol{B}, \tag{2}$$

Please note that in many publications this equation is written with the $+\boldsymbol{u} \times \boldsymbol{B}$ term on the $\boldsymbol{E}$ side. Often $\boldsymbol{E}$ is
measured in some frame (for example the Earth-fixed one), and the task is to estimate $\boldsymbol{E}^*$. In this work we prefer to write the relation as in Equation 2. In the top ionosphere and plasmasphere collisions are rare and the plasma drifts such that $\boldsymbol{E} + \boldsymbol{v} \times \boldsymbol{B} = 0$, $\boldsymbol{v}$ the ion or electron drift. This means that the electric field in the frame of the plasma is zero, and also the cross-B current. The conductivity along $\boldsymbol{B}$ is very high compared to the Pedersen and Hall conductivities, and in the steady state $E_\parallel = 0$. For constant $\boldsymbol{B}$ then also $\boldsymbol{E}(z)$, $z$ the coordinate along $\boldsymbol{B}$, is
constant. This justifies using height-integrated quantities in Equation 1. When comparing electric fields between the magnetosphere and ionosphere, $\boldsymbol{B}$ is not constant and $\boldsymbol{E}$ is said to "map" between positions along $z$ (Kelley, 2009, Chapter 2.4). For the scenario in Figure 1 we also request such mapping of $\boldsymbol{E}$ between hemispheres. Owing to the highly symmetric preconditions the mapping is simply that a northward $\boldsymbol{E}$ in the northern hemisphere maps to southward in the southern hemisphere with equal magnitudes, and analogous for reversed directions of $\boldsymbol{E}$.
The Pedersen current driven by $\boldsymbol{E}^*$ is associated with Joule or frictional heating (Vasyliūnas and Song, 2005) with power in Watts per m$^2$

$$Q_J = \Sigma_P \boldsymbol{J}_P \cdot \boldsymbol{E}^* = \Sigma_P (\boldsymbol{E}^*)^2 = (\boldsymbol{J}_P)^2 / \Sigma_P \tag{3}$$




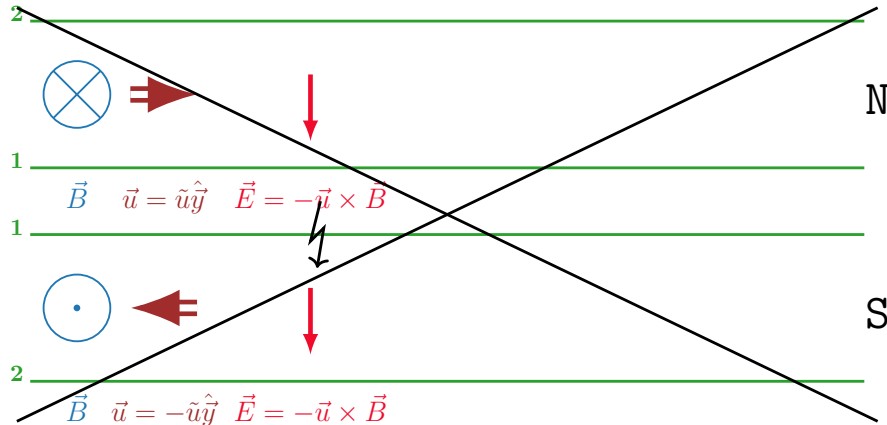

**Figure 2.** Collapsed 2-d view of the scenario sketched in Figure 1 with the northern and southern regions nearly adjacent to each other and as seen from above the ionospheric dynamo region. The latitude circles labeled "1" and "2" are magnetically connected, respectively. The lightning bolt symbolizes an electrical short circuit along $\boldsymbol{B}$ by electrons that would occur for the suggested $\boldsymbol{E}$. The large black cross indicates that this scenario is rejected as a possible electric field configuration, please see the text.

The divergent $\boldsymbol{J}_P$ connects to field-aligned currents (FACs). These currents are associated with a magnetic perturbation $\Delta B = \mu_0 \Sigma_P E^*$ in the top ionosphere (Sugiura, 1984). The difference of the Poynting flux above and below the dynamo region is $\boldsymbol{E}^* \times \Delta \boldsymbol{B}/\mu_0$ matching $Q_J$ in Equation 3. For the sake of brevity we say that the Poynting flux is downward and equal to the Joule heating rate, for $\boldsymbol{E}$ in the frame of neutral gas, where $\boldsymbol{E} = \boldsymbol{E}^*$.

## 3  Symmetric Dynamos

Our aim is to figure out the correct electric field configuration for the scenario sketched in Figure 1. For this a collapsed 2-d view of the 3-d one, with the northern stripe of zonal neutral wind just above the southern one, is useful. Both regions are viewed from above the dynamo region, respectively. The view is shown in Figures 2-4. In a first attempt we consider the reference frame fixed to the Earth and assume that $\boldsymbol{E}^* = 0$. There are still electric
fields as a result of the neutral winds in both hemispheres according to Equation 2, $\boldsymbol{E} = -\boldsymbol{u} \times \boldsymbol{B}$. This first try is sketched in Figure 2. In both, the northern part $N$ and the southern one $S$, $\boldsymbol{E}$ points southward, because both $\boldsymbol{u}$ and $\boldsymbol{B}$ change to opposite directions. But this configuration of $\boldsymbol{E}$ implies a potential drop along magnetic field lines connecting either "1" or "2" or both. Electrons would short circuit such potential drops. Instead, the plasma will establish an electric field $\boldsymbol{E}^*$ (perpendicular to $\boldsymbol{B}$) including an $\boldsymbol{E}^* \times \boldsymbol{B}$ drift in the plasmapher, such that
potentials along $\boldsymbol{B}$ are avoided. We therefore reject the initial idea that the only electric fields are from of Galilei coordinate transformations from neutral to observer frames.





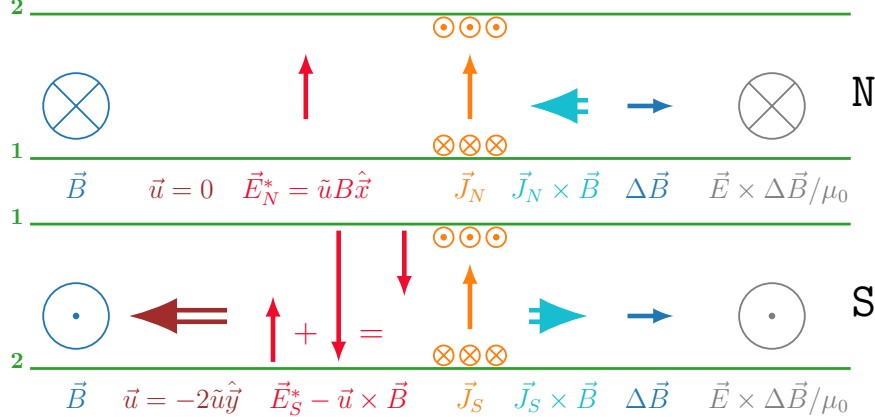

**Figure 3.** 2-d view like in Figure 2, but in the reference of the northern neutral gas. Electric fields $E_N^*$ and $E_S^*$ avoiding potential drops along $\boldsymbol{B}$ are now added, obtaining consistent electric current $\boldsymbol{J}$, $\boldsymbol{J} \times \boldsymbol{B}$ force, magnetic stress $\Delta \boldsymbol{B}$ and Poynting flux $\boldsymbol{E} \times \Delta \boldsymbol{B}/\mu_0$.

Now we attempt to find a consistent configuration such that $\boldsymbol{E^*} \neq 0$, i. e. a non-zero electric field in the neutral frame. Figure 3 shows the result in the same format as Figure 2, but in a reference frame where the northern neutral wind is zero, and consequently the southern easterly wind twice as strong (both $N$ and $S$ are always shown in the same reference frame). Guided by the highly symmetric preconditions we guess that $E_N^*$ has to point northward

with magnitude $\tilde{u}B$, with the strength of the wind being $\tilde{u}$. Equations by which $E^*$ can be determined instead of guessed are given in the following section.

$E_N^*$ drives a northward current $J_N$, resulting in an westward $\boldsymbol{J_N} \times \boldsymbol{B}$ force. $J_N$ connects to FACs at the edges of the neutral wind jet, where we assume that $\boldsymbol{u}$ and therewith $\boldsymbol{E^*}$ drop to zero. The magnetic stress $\Delta B$ from the current is eastward, from which we derive a downward Poynting flux $\boldsymbol{E_N} \times \Delta \boldsymbol{B}/\mu_0$ matching the Joule heating

$J_N \cdot E_N^*$ in the $N$ dynamo.

In $S$ $E_S^* = \tilde{u}B$ is also northward, i. e. $E_N^*$ and $E_S^*$ *do not map* to each other along the magnetic field. Being still in the frame of the northern neutral gas, the southward electric field from the westerly neutral wind $-2\boldsymbol{u} \times \boldsymbol{B}$ is added and gives the total $\boldsymbol{E}$ in $S$. This field *does map* to $E_N^*$ at the conjugate point, i. e. magnetic field lines are equi-potentials. The current $\boldsymbol{J_S}$ is driven by the electric field in a zero neutral wind reference frame, which is, for $S$,

$\boldsymbol{E_S^*}$. $\boldsymbol{J_S}$ correctly closes the current loop between $N$ and $S$, such that $\nabla \cdot \boldsymbol{j} = 0$. $(\boldsymbol{E_S^*} - \boldsymbol{u} \times \boldsymbol{B}) \cdot \boldsymbol{J_S} < 0$ which suggests that $S$ is a "dynamo". This and an upward Poynting flux $\boldsymbol{E_S} \times \Delta \boldsymbol{B}/\mu_0$ is consistent with the notion, that the $S$ dynamo drives Joule heating in $N$ via Poynting flux from $S$ to $N$.

To fully assert consistency, Figure 4 shows the 2-d scene in the reference frame where the southern neutral wind is zero. Currents, forces, and magnetic stress are invariant under Galilei transformation and do not change. The total

electric field and the Poynting flux are not invariant and so change compared to Figure 3. It becomes clear that



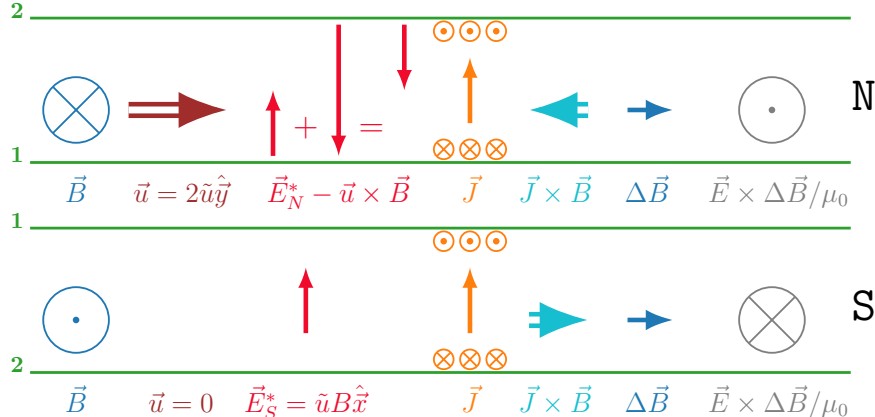

**Figure 4.** 2-d view like in Figure 3, but in the reference of the southern neutral gas. Electric fields $E_N^*$ and $E_S^*$ that are set up by the plasma to avoid $E_\parallel \neq 0$ are the same, but the $\boldsymbol{u} \times \boldsymbol{B}$ field as a result of the Galilei transformation from the neutral wind frame to the observer is now in the northern hemisphere. Please see the text for further discussions.

Joule heating also occurs in $S$, driven by the dynamo in $N$ via Poynting flux from $N$ to $S$. The magnitude in both hemispheres is the same because of the assumed symmetric preconditions.

The title of this section "symmetric dynamos" does not refer to in an Earth fixed frame symmetrically opposing zonal winds as drawn in Figure 1. The same results are obtained for equal wind difference but an asymmetric zonal wind specification. "Symmetric" rather refers to equal ionospheric conditions at the conjugate points, equal magnetic field strengths and perfectly opposing field directions, which is partially surrendered in the following section.

## 4  Asymmetric Dynamos

The strengths of currents and forces, and the magnitudes of Joule heating and Poynting fluxes are proportional to the Pedersen conductance $\Sigma_P$. To edge the model a little bit towards a more realistic one, we allow now for different values $\Sigma_N$ and $\Sigma_S$ in each hemisphere. An obvious motivation is, that near solstices one dynamo might be sunlit while the conjugate one is not. In addition, considering the asymmetric case provides an opportunity to write down equations for the fields $\boldsymbol{E}_N^*$ and $\boldsymbol{E}_S^*$ instead of guessing them.

1. the total electric fields in $N$ and $S$ using the same reference frame has to map (avoiding a non-zero $E_\parallel$). We choose arbitrarily the frame of the northern neutral gas (Figure 3):

$$E_N^* = E_S^* + \Delta u B \tag{4}$$

for given zonal wind difference $\Delta u = 2\tilde{u}$. $\Delta u$ is positive for $u_{y,N} > u_{y,S}$.



2. the current loop between $N$ and $S$ close exactly. In each $N$ and $S$ the current is determined by the electric field in the reference frame of zero neutral wind, which are $\boldsymbol{E}_N^*$ and $\boldsymbol{E}_S^*$, respectively. So for the current calculation the frames in $N$ and $S$ are not the same:

$$\Sigma_N E_N^* + \Sigma_S E_S^* = 0 \tag{5}$$

The solutions of equations 4 and 5 are

$$E_N^* = \frac{\Sigma_S}{\Sigma_N + \Sigma_S} \Delta u B = -\frac{\Sigma_S}{\Sigma_N} E_S^* \tag{6}$$

and

$$E_S^* = -\frac{\Sigma_N}{\Sigma_N + \Sigma_S} \Delta u B = -\frac{\Sigma_N}{\Sigma_S} E_N^* \tag{7}$$

The Pedersen current is the same in both hemispheres:

$$J = \frac{\Sigma_N \Sigma_S}{\Sigma_N + \Sigma_S} \Delta u B \tag{8}$$

Figure 5 shows how $\boldsymbol{E}^*$ gets adjusted in a situation where $\Sigma_N = 0.5$ S (or mho) and $\Sigma_S = 1.0$ S. The values are perhaps realistically a bit low, and could be more different. They were chosen so that the lengths of vectors $\boldsymbol{E}$ in Vm$^{-1}$ and $\boldsymbol{J}$ in Am$^{-1}$ have the same scale in the Figures, for better visual understanding. The reference frame is that of the northern neutral gas. The lower $\Sigma_N$ implies a larger $E_N^*$ compared to the values in $S$, and also implies stronger Joule heating which is supplied by a higher Poynting flux from $S$ to $N$. Please compare with Figure 6 showing the same scenario, but in the frame of the southern neutral gas.

The Joule heating in each hemisphere is

$$Q_N = \Sigma_N \left( \frac{\Sigma_S}{\Sigma_N + \Sigma_S} \Delta u B \right)^2 = \frac{\Sigma_S}{\Sigma_N} Q_S \tag{9}$$

and

$$Q_S = \Sigma_S \left( \frac{\Sigma_N}{\Sigma_N + \Sigma_S} \Delta u B \right)^2 = \frac{\Sigma_N}{\Sigma_S} Q_N \tag{10}$$

and the total Joule heating

$$Q = Q_N + Q_S = \frac{\Sigma_N \Sigma_S}{\Sigma_N + \Sigma_S} (\Delta u B)^2 \tag{11}$$

A low $\Sigma_P$ in either $N$ or $S$ reduces both the current and the total Joule heating.

## 5 Discussion

A current system connecting both hemispheres has been suggested first by van Sabben (1966) for Sq. Evidence for interhemispheric field-aligned currents were presented first by Olsen (1997) with Magsat. Paraphrasing Yamazaki





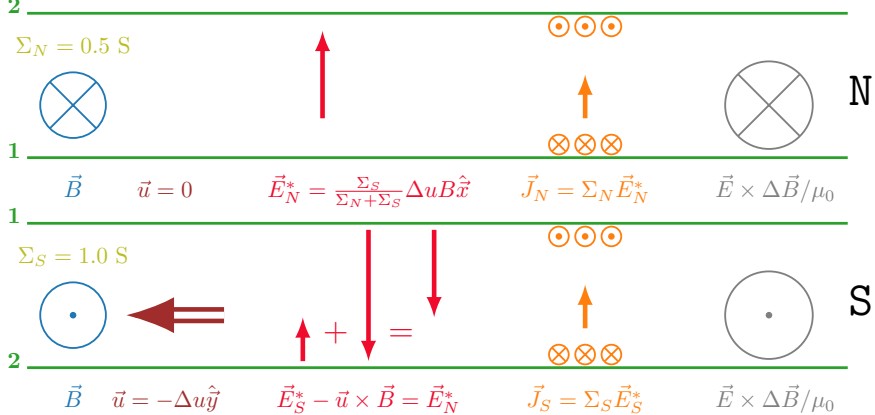

**Figure 5.** 2-d view like in Figure 3, in the reference frame of the northern neutral gas. The electric fields are such that for asymmetric conductances $\Sigma_N = 0.5$ and $\Sigma_S = 1.0$ the same current $\boldsymbol{J}$ is obtained. $\boldsymbol{J} \times \boldsymbol{B}$ force and magnetic stress $\Delta \boldsymbol{B}$ are omitted in this Figure. The sizes of the symbols for Poynting flux in Figure 6 and this Figure are according to the flux magnitudes having the same scale.

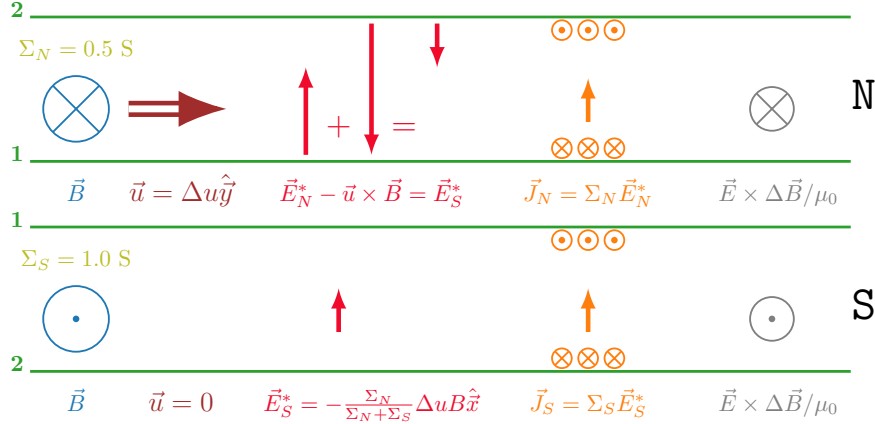

**Figure 6.** 2-d view like in Figure 4, in the reference of the southern neutral gas. The electric fields are such that for asymmetric conductances $\Sigma_N = 0.5$ and $\Sigma_S = 1.0$ the same current $\boldsymbol{J}$ is obtained. $\boldsymbol{J} \times \boldsymbol{B}$ force and magnetic stress $\Delta \boldsymbol{B}$ are omitted in this Figure. The sizes of the symbols for Poynting flux in Figure 5 and this Figure are according to the flux magnitudes having the same scale.

and Maute (2017), similar analysis was later performed with Oerstedt, Champ and Swarm satellite data, see references therein. Arguing with charge transport from one hemisphere to the other, already Fukushima (1979) had suggested that there are electric potential differences between conjugate points. Presumably such potential differ-





ences implicitly exist also in global circulation models (GCMs) that include the thermosphere. A typical code will maintain specified relations between spatially neighboring and nearby grid points. But unless special care is taken a potential difference should develop between magnetically conjugate grid points which are normally not nearby to each other in a global simulation.

Here we have started by figuring out the electric field configuration for magnetically connected regions in different hemispheres, using a highly symmetric configuration with zonal winds only. Noting that field-aligned potentials would be the result if exclusively the condition $\boldsymbol{E} + \boldsymbol{u} \times \boldsymbol{B} = 0$ determined $\boldsymbol{E}$, we have rejected this possibility and instead sought a solution where plasma drifts avoid this. The drifts are associated with non-zero $\boldsymbol{E}^*$ in the reference frame of the local neutral gas.

The obtained solution automatically depends only on relative wind differences along $\boldsymbol{B}$. A wind without any variations along $\boldsymbol{B}$ would not force the plasma to establish an $\boldsymbol{E}^*$, it then cannot drive currents, the electric field in the neutral frame being zero, and it does not drive any dynamo. Vasyliūnas (2012) made a similar point. For entangled dynamos the only frames relevant are connected to locally interacting matter, namely the neutral gas (and as discussed below, the plasma). Irrelevant are aether-like absolute reference frames, as the Earth-fixed frame would

be one in this context. We consider this a good agreement with fundamental physical principles.

Recently Khurana et al. (2018) and Provan et al. (2019) interpreted magnetic features during the final passes of the Cassini spacecraft in terms of neutral gas velocity differences in Saturn's upper atmosphere. There are similarities between their interpretations of the Cassini data and our model for the Earth's Sq. In the Jovian magnetosphere there is magnetic conjugacy between the ionospheres of the Io moon and its footprint in Jupiter's atmosphere which

also leads to a quasi-staedy state current system connecting both regions (Huang and Hill, 1989). Entangled Joule heating as a consequence of the systems at Jupiter and Saturn had not been addressed in these studies, but should in principle occur also there.

The main features of the entangled dynamo model are

1. avoidance of field-aligned potential drops,

2. and dependence only on relative motion, no reference to absolute frames.

The current system of entangled dynamos is qualitatively similar to van Sabben's. Compared to a model or description that allowed for field-aligned potential differences and or or depended, possibly implicitely, on winds in an absolute Earth fixed frame, there would be quantitative differences and divergence in details. Satellite data have not been analysed and simulations performed with such differences in mind.

There are complications that will need to be taken into account for a quantitative comparison with data or simulations, among them

– the tilt of the geomagnetic field with respect to geocentric or geodetic coordinates, which are the natural ones for the neutral wind;



- other deviations of the geomagnetic field from a centered dipole;

- the non-vertical inclination of $\boldsymbol{B}$ in the dynamo layers;

- meridional winds;

- the interdynamo current (equation 8) is the same in both hemispheres, a low $\Sigma_P$ in one hemisphere being
 compensated for by a stronger $E^*$, and vice versa. However, the Sq ground variations are from Hall currents.
 If the ratio $\Sigma_H/\Sigma_P$ is different between the conjugate points, then the Sq variations are also asymmetric.

This work focuses on a fundamental understanding of the ionospheric dynamo interactions. The construction of a more realistic model based on entangled dynamos and detailed comparison with data should be relatively straight forward, but here it is an outlook for the future.

Nevertheless we find it implausible that the plasma would support potential differences between hemispheres over large scales and long times. Therefore we argue, that the Sq phenomenon is essentially driven by *wind differences* at conjugate points, and entangled dynamos are a convenien way to describe the mechanism. Interhemispheric wind differences do obviously come about when atmospheric circulation and tides are not symmetric with respect to the equator. This asymmetry maximizes near solstices. But probably more decisive factors are the tilt of the geomagnetic

field's dipole axis, its offset from the Earth centre, and deviations from a with respect to the dipole equator perfectly symmetric field. These cause wind differences also near equinoxes when the wind pattern itself should be relatively symmetric with respect to the geodetic equator. The relatively strong semi-diurnal component of the Sq variations at the ground is consistent with a misaligned rotator being involved, as a misalignment tends to produce signals at half the rotation period. The neutral wind pattern itself in geodetic coordinates would not necessarily have a semi-

diurnal component. A semi-diurnal component can get excited in the neutral wind itself by the dynamo interactions if the forcing of the thermosphere by the plasma is sufficiently effective. Other explanations for the semi-diurnal component in Sq have been given (Yamazaki and Maute, 2017).

The Sq variations at the ground are by Hall currents, which is of course well-known and accepted. However, an $E^*$ must exist to drive the interdynamo currents (equation 8) as well as any Hall currents. A non-zero $E^*$ is for

example created if the local thermospheric wind is zero relative to the observatory, but strong at the conjugate point. No effect is observed, if there is a strong local thermospheric wind, and the same strong wind at the conjugate point. The local thermospheric wind relative to the observer *alone* has no significance for the entangled dynamo mechanism.

In sections 2–4 we have depicted the wind difference as jet-like, in order to achieve a good insight into the

entanglement of dynamos. According to ground magnetometer observations Sq is a counter-clockwise current vortex in the northern hemisphere covering the entire dayside, and a clockwise vortex in the southern hemisphere (Yamazaki and Maute, 2017). The actual wind difference $\Delta\boldsymbol{u}$ is therefore not jet-like, but also vortex-like, with opposite polarity as the current. The neutral wind in geodetic coordinates may only little resemble these vortices, because magnetic



ground observations provide a heavily filtered and transformed image of it: The modulation of the conductances $\Sigma_P$ and $\Sigma_H$ by solar ionizing radiation creates a strong diurnal component, and the misaligned near-dipolar geomagnetic field cartographically maps wind differences non-linearly and in a skewed way from the geodetic coordinate system. This mapping varies with longitude. The longitudinal dependence is indeed seen in the FAC pattern, please confirm for example with Olsen (1997) and Park et al. (2011). The spatial sparsity of ground observatories make it difficult to simultaneously extract both diurnal and longitudinal components, while this is possible using LEO satellite data.

In summary, particularly the semi-diurnal component and the simultaneous LT and longitudinal dependence are observational characteristics that are consistent with a driver of entangled dynamos and we argue that both of these features naturally arise from it.

A striking feature of entangled dynamos is the kinetic energy extracted from the neutral wind at one dynamo and dissipated as Joule heating at the other dynamo. We obtain a rough estimate of the total Joule heating $Q_{J,hem}$ using

$$Q_{J,hem} \approx \frac{J_{P,tot}^2}{\langle\Sigma_P\rangle} \approx \left(\frac{\langle\Sigma_P\rangle}{\langle\Sigma_H\rangle}\right)^2 \frac{J_{H,tot}^2}{\langle\Sigma_P\rangle}, \tag{12}$$

where $J_{P,tot}$ $J_{H,tot}$ are the total horizontally integrated Pedersen and Hall currents in Ampere, respectively, and $\langle\Sigma_P\rangle$ and $\langle\Sigma_H\rangle$ are average dayside values for the Pedersen and Hall conductances, respectively. Takeda (2015) determined $J_{H,tot}$ from observatory data to quite consistently between about 100 and 200 kA. Ieda et al. (2014) investigated the solar zenith angle $\chi$ dependence of $\Sigma_P$ and $\Sigma_H$ at quiet times, i. e. without interference from particle precipitation. Adopting an "average" dayside $\chi$ of 25° we extract $\langle\Sigma_H\rangle/\langle\Sigma_P\rangle \approx 1.4$ and $\langle\Sigma_P\rangle \approx 9$ S from Ieda et al. (2014). Consequently a global estimated Joule heating power of Sq due to wind differences between conjugate points of roughly *between 0.5 and 2 GW per hemisphere* is obtained.

Generally accepted is the importance of Joule heating at high latitudes where it varies very strongly with geomagnetic activity. The high-latitude Joule heating power has mainly been estimated for major geomagnetic storms. Using EISCAT radar measurements during the Halloween storm in 2003 to scale an AMIE data assimilation (Richmond and Kamide, 1988), Rosenqvist et al. (2006) estimated the Joule heating power in the high-latitude northern hemisphere to about 2.4 TW, exceeding our estimate for Sq by three orders of magnitude. Somewhat lower peak values of about 1 TW were obtained in numerical simulations (e. g. Fedrizzi et al., 2012). However, such peak values are obtained only for times of an hour or so. The average Joule heating in storm periods would be much lower, perhaps by a factor of ten or so. Most common is actually low geomagnetic activity, when the amount of high-latiude Joule heating is poorly known. The neutral dynamo driven Joule heating is a permanent, relatively constant trickle, which when integrated sufficiently long times, i. e. several solar rotations, may well turn out to be significant compared to the high-latitude Joule heating. The Joule heating from Sq is buried in heating from ionizing solar radiation. To compare the order of magnitude of both heating processes, we assume that an ionization and recombination consumes about 35 eV$= 5.6\cdot10^{-18}$ J (Rees, 1989). This value stems from laboratory mesurements of ionization by electron impact and is often used to estimate heating by electron precipitation in the aurora. We assume that it





roughly applies also in case of ionization by solar radiation in the E region. The coefficient of dissociatve recombination $\alpha \approx 3.5 \cdot 10^{-13}$ m$^3$s$^{-1}$ (Bates, 1988). We compare the heating within a layer of $\Delta z = 10$ km centered at the peak of $\sigma_P(z)$, which is typically at about 130 km altitude in the E region. Then the heating by solar radiation

$$Q_S \approx 5.6 \cdot 10^{-18} \alpha \langle N_e \rangle^2 \Delta z \tag{13}$$

in Wm$^{-2}$. The heating in one dayside hemisphere

$$Q_{S,hem} = \pi R_E^2 Q_S \tag{14}$$

with $R_E = 6378$ km the Earth radius. It remains to give a representative value for the electron density $\langle N_e \rangle$ matching $\langle \Sigma_P \rangle \approx 9$ S, which was used above the estimate the Sq Joule heating. To simplify the integration over height we use instead

$$\langle \Sigma_P \rangle \approx \Delta z \frac{e \langle N_e \rangle}{2B} \tag{15}$$

with $B = 35000$ nT as an average value of the magnetic field strength at mid latitudes. This gives $\langle N_e \rangle \approx 4 \cdot 10^{11}$ m$^{-3}$ and a solar heating per hemisphere of $Q_{S,hem} \approx 400$ GW. Clearly this is only a very rough estimate and does particularly not take into account any solar cycle variations that are certainly present. These would affect both the heating by solar radiation and by the neutral dynamos (Sq). For now we tentatively state that *the Joule heating by*

*Sq amounts to roughly 0.1-1 % of the solar heating in the same altitude range*, with a more quantitative investigation as an outlook for the future.

  We claim that there is Poynting flux from $N$ to $S$ as well as from $S$ to $N$, each transporting electrodynamic energy from a dynamo to a load. Adding both Poynting fluxes would give zero (in the symmetric case), but this is not a meaningful view. The Poynting flux $\boldsymbol{E} \times \Delta \boldsymbol{B}/\mu_0$ as well as the term $\boldsymbol{J} \cdot \boldsymbol{E}$ depend on the reference frame.

There are infinitely many possible reference frames, and in each of these Poynting's theorem is of course valid. But only frames of zero neutral wind are special, with the $\boldsymbol{J} \cdot \boldsymbol{E}^*$ term and the ionospheric Ohm's law giving the dissipation. We argue that it is in this frame where $\boldsymbol{J} \cdot \boldsymbol{E}$ represents the neutral dynamo's power in Wm$^{-2}$ and the Poynting flux the amount and direction of electromagnetic energy being transported from the dynamo to the load. On each magnetic flux tube the neutral winds at each conjugate end define so two reference frames connected to

physical material. In each of the two frames one end is the location of the load. At the other end is a dynamo where $\boldsymbol{J} \cdot \boldsymbol{E} = \boldsymbol{J} \cdot (\boldsymbol{E}^* - \Delta \boldsymbol{u} \times \boldsymbol{B}) < 0$ matching the dissipation at the load. When switching the reference frames the roles also switch, and the Poynting flux between both ends flips to the opposite direction. The neutral dynamo power is so determined by the neutral wind difference at the conjugate points.

  After some hesitation about adding an in the field of ionospheric physics new concept, we have nevertheless adopted

the adjective "entangled" as a being descriptive and concise. Entangled is originally and widely used for quantum mechanical states. As far as we are aware of, in classical physics "entangled" is nowhere prominently used, and a mixup therefore unlikely. The German "verschränkt", used originally by Schrödinger (1935) (see also Trimmer, 1980)





for these quantum mechanical states, describes the situation also for conjugate ionospheric dynamos in a linguistic sense especially well. An observer experiences "action at a distance" in that wind variations far away at the conjugate point control the local currents, electric field, and Joule heating. Such "action at a distance" is of course normal in classical current circuits. Wind changes are communicated with a tiny delay given by the Alfvén velocity through
the plasmasphere. In a practical sense it is instanteneous considering how slowly the neutral wind typically changes.

Vasyliūnas (2012) concluded that steady state dynamo currents exist 1) only for a neutral wind with gradients (more precisely, if $\nabla \times (\boldsymbol{u} \times \boldsymbol{B}) \neq 0$), or 2) if boundary conditions above the dynamo region impose a non-zero current. In our entangled dynamo model gradients of the neutral wind within each dynamo were neglected, or assumed to be zero. Apparently the model belongs to Vasilyūnas' second category of possible dynamos, with the conditions in
each hemisphere determining the boundary conditions at the other hemisphere. The model of entangled dynamos avoids specifying the shallow transition between $\boldsymbol{E}_N^*$ and $\boldsymbol{E}_S^*$ along $z$ that must occur in the plasmasphere including a transition between the corresponding plasma drifts. We consider this as an advantage. Simulations and models of the neutral atmosphere normally don't extend into the plasmasphere, and most available data are from the ionosphere. However, in principle the transition would be determined by the transition of the neutral wind between hemispheres
through the plasmasphere according to

$$E^*(z) + \Delta u(z) B(z) = \mathrm{const}, \tag{16}$$

where the coordinate $z$ along $\boldsymbol{B}$ is from the bottom of the dynamo region in $S$ to that in $N$. Even though the interaction between neutral gas and plasma becomes weak, the neutral atmosphere does extend into the plasmasphere. It is of course possible the describe the entire system as one, and then the prescribed $\boldsymbol{u}$, confirm Figure 1,
has apparently $\nabla \times \boldsymbol{u} \neq 0$ fulfilling the demand of Vasyliunas' first category. But we see advantages for the concept of separate, entangled dynamos, as it efficiently focuses on the regions of strong dynamo and heating effects.

Applying the entangled dynamo model in modified form to high-latitudes turns out to be instructive as well: if the magnetically connected counterpart of the neutral atmosphere is an "active" plasma in space, without a conjugate point in the opposite hemisphere, then this describes "ordinary" ionosphere-magnetosphere coupling. The "active"
plasma acts as dynamo, the neutral atmosphere is the load, consistent with the established paradigms. A sketch how particularly the solar wind can act as dynamo is for example in Rosenqvist et al. (2006). The appropriate reference frame for the steady state is that of the neutral gas, the load. It is well accepted that high latitude electric fields, Poynting flux and Joule heating need to be calculated in this reference frame. So far nothing special about ionosphere-magnetosphere coupling has emerged. However, looking for an entangled counterpart, we find that it is
not present: the "partner" candidate for a load would be the plasma. But in the reference frame of the plasma $\boldsymbol{E} = 0$, the plasma does not dissipate energy, and also the Poynting flux in the plasma reference frame is zero. Thus it seems very doubtful that the neutral wind can act as a dynamo on open field-lines when the plasma in space is the only possible load. From ion-electron collisions there would be only a very tiny, insignificant non-zero electric field in the





plasma frame. When treating the space plasma as ideal (= without collisions), a dynamo works only in the direction from space to the Earth's atmosphere, as opposed to a system with neutral dynamos at both ends.

Measurements of the electric field with satellites are normally transformed to the Earth fixed frame,

$$\boldsymbol{E}_e = \boldsymbol{E}_s + \boldsymbol{v}_{orb} \times \boldsymbol{B}, \tag{17}$$

with $\boldsymbol{v}_{orb}$ the satellite velocity in the Earth fixed frame, $\boldsymbol{E}_{orb}$ the observed or measured electric field, and $\boldsymbol{E}_e$ the transformed one. Desired is really

$$\boldsymbol{E}_n = \boldsymbol{E}_s + (\boldsymbol{v}_{orb} + \boldsymbol{u}) \times \boldsymbol{B} \tag{18}$$

in the neutral gas frame, however, the neutral wind is unknown, or known only with large uncertainty. At high latitudes plasma drifts in the Earth fixed frame are typically much larger than the neutral wind. Therefore, Poynting flux and Joule heating may approximately be estimated in the Earth fixed frame instead of the neutral frame. A large amount of satellite data have so been processed, resulting in average spatial patterns of Joule heating and downward Poynting flux in the polar ionosphere (Gary et al., 1994; Waters et al., 2004), which are undisputably very valuable and relevant results. In certain areas of open field-lines, however, consistently a weak upward Poynting flux is obtained. This merely indicates that in these areas there is a consistent non-zero neutral wind and also relatively weak plasma drift. The Poynting flux evaluated in the Earth fixed frame, which is the "wrong" frame, can then turn out upward. In light of the discussion in the previous paragraph it seems doubtful that the neutral gas can act as dynamo for the collisionless plasma in space on average over larger areas. To prove such a possible neutral wind dynamo against the space plasma in satellite data, the observed $\boldsymbol{E}_s$ would need to be transformed into the plasma reference frame using ion drift measurements from the satellite. The expected outcome is, within measurement uncertainties, zero. The neutral wind would not act in any significant way as a dynamo against the space plasma.

## 6 Conclusions and Outlook

Considered is the situation of a "passive plasma", i. e. where the electric field in the reference frame of the neutral gas is zero, and a neutral wind that is not constant along the magnetic field. From the paradigm that the plasma in the steady state avoids electric potential differences along $\boldsymbol{B}$ we conclude that the plasma then cannot remain "passive" and will be drifting with associated electrostatic $\boldsymbol{E}^*$ perpendicular to $\boldsymbol{B}$, so preventing any $E_{\parallel} \neq 0$. $\boldsymbol{E}^*$ drives currents fullfilling $\nabla \cdot \boldsymbol{j} = 0$ and Joule heating with $\boldsymbol{j} \cdot \boldsymbol{E}^* > 0$. $\boldsymbol{E}^*$ is not constant along $\boldsymbol{B}$, if $\boldsymbol{u}(z)$ varies.

We have particularly looked at the situation of two regions of interacting neutral gas and plasma in opposite hemispheres, with a dipole-like $\boldsymbol{B}$ connecting both regions and defining conjugacy. Then any difference between the neutral winds at conjugate points results in a dynamo effect. Electromagnetic energy is generated and transported to the opposite hemisphere as Poynting flux. There the energy is dominantly dissipated, due to a large mass of the





neutral gas compared to the plasma. The process works in both directions, and entanglement seems a convenient and useful description of the situation. We agree with Vasyliūnas (2012) and others, that an isolated neutral wind in a plasma would not result in any steady state dynamo effect.

We suggest that the Earth's magnetic Sq variations are driven by neutral wind differences at conjugate points. The main dipole geomagnetic field is tilted with respect to the Earth's rotation axis as well as it is not centered, making it a strongly misaligned rotator. This would explain the presence of a 12-hour component in Sq variations. The dynamo currents are modulated by the product of the Pedersen conductances in both hemispheres resulting also in a 24 hour component of the variations at a fixed point on the Earth. In addition the Sq variations reflect of course also dynamics of the neutral atmosphere itself, in as far as it involves wind differences at conjugate points.

The Joule heating driven by the neutral wind in the Earth's thermosphere is estimated to about 0.5 to 2 GW per hemisphere, quasi-permanently moving around the Earth with the Sun. According to a rough estimate of the order of magnitudes this Joule heating is about 0.1 to 1 percent of the energy consumed by ionization from solar radiation and its recombination in the same altitude range. The Joule heating by Sq is near constant compared to the high-latitude Joule heating which varies over several orders of magnitude depending on geomagnetic activity.

The prescriptions for obtaining the electrostatic field, stationary drifts and currents in the space plasma interacting with a neutral atmosphere are in general terms:

1. potential differences are avoided along $\boldsymbol{B}$ and the electric field maps;

2. field-aligned currents close across $\boldsymbol{B}$.

These are already well accepted principles in ionosphere-magnetosphere coupling in the steady state (Kelley, 2009,
Chapter 2). Applying the prescriptions to the situation of a neutral wind that is not constant along a coordinate $z$ along $\boldsymbol{B}$ has helped us to clarify that

1. the mapping electric field is $\boldsymbol{E}^* - \boldsymbol{u} \times \boldsymbol{B}$ and the mapping needs to be done in one single reference frame. The neutral gas does not define the frame unambiguously as the wind varies along $z$. The frame for the mapping can be choosen freely, but it must be the same frame all along $z$;

2. the closure current is $\sigma_P \boldsymbol{E}^*$, and it is evaluated in the reference frame of the local neutral gas, such that the frame relevant for current calculation generally changes along $z$.

We have not been able to present direct empirical evidence that the entangled dynamo model is the correct one for Sq. In numerous previous works a dynamo effect had, often implicitly, been attributed to a neutral wind per se (in a fixed frame, like the Earth's). A detailed analysis of existing and future data from satellites and the ground
with respect to the entangled dynamo model and neutral wind differences along $\boldsymbol{B}$ is left as a future task.

This prescription above has the potential to guide the design of methods for simulating atmospheric and plasma circulation, such that explicitly potential drops between opposite hemispheres are avoided. We believe that this approach applied in GCMs would be realistic and may give significantly different plasma convection and neutral



wind compared to methods which are currently in use and which do not enforce zero potential difference at conjugate points. AMIE-like data assimilation for estimating neutral wind differences between hemispheres from observations of FACs, for example with the Swarm mission, and groundbased magnetic data seems possible. Such neutral wind differences would then usefully constrain estimates of the absolute wind by other methods.

The concept of two-way entangled dynamos is applicable for the mid-latitudes and Sq, but not on open field lines and only with modifications at equatorial latitudes. For high latitudes we have briefly discussed the concept of the plasma in space acting as a dynamo driving Joule heating in the thermosphere (but not vice versa), which is just an alternative phrasing of well established concepts of ionosphere-magnetosphere coupling, applicable on open magnetic field-lines.

On closed field-lines the currents and fields of entangled dynamos can coexist with currents and fields induced by plasma motion in the magnetosphere driven by interaction with the solar wind, to use a generic term. This includes sub-storms, high-latitude plasma convection, its occasional penetration towards lower latitudes etc.

    Near the equator phenomena are more complex again, as is well known. The here presented dual entangled model seems to have limited applicability for the equatorial ionosphere. In quantum mechanics entanglement is not

restricted to dual (Greenberger et al., 1989). Entanglement of the dynamos in the equatorial F and E regions might turn out to be an applicable concept.

*Author contributions.* Stephan Buchert did the research and wrote the article.

*Competing interests.* No competing interests are present

*Acknowledgements.* I thank Profs. Gerhard Haerendel and Ryo Fujii for fruitful discussions, and also David Andrews for
pointing me to the Cassini results.





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
