# Peer review of "Entangled Dynamos and Joule Heating in the Earth's Ionosphere"

_Annales Geophysicae, 2019_

## Author Comment (AC1) · 23 Aug 2019

**Author comment on**
**Entangled Dynamos and Joule Heating in the Earth's Ionosphere**

Stephan C. Buchert[1]

[1]Swedish Institute of Space Physics, Uppsala, Sweden

**Correspondence:** Stephan Buchert (scb@irfu.se)

**Abstract.**

In this commment I would like to point out a few errors in the manuscript, that in my opinion are minor and do not affect the overall content and conclusions. I apologize for not having noticed all errors and removed them before submission.

Secondly I provide clarifications and statements which are hopefully helpful. They are not additional new conclusions and do not substantially change the conclusions of the manuscript.

**1 Correction of Errors and Other Amendments**

– Page 3, line 8: "..., and also the cross-B current." will be erased. Currents across field lines in the topside ionosphere and plasmasphere are not relevant in this context, they cannot be excluded, and adding this statement was not justified.

– Page 5, line 7: "... in an westward ..." should be "... in a westward ..."

– In Figure 6 of the manuscript the Poynting flux is erraneously plotted with the wrong polarity. In the reference frame of the southern neutral gas the Poynting flux is into $S$ and out of $N$. The corrected Figure is shown below.

– the work by Fukushima (1979) is misrepresented in the manuscript. Fukushima theoretically estimated the field-aligned conductivity and so calculated potential differences of a few volts between hemispheres for $Sq$ FACs that had been observed with satellites. Such low voltages are consistent with entangled dynamos. The potential differences arising from uncompensated $\boldsymbol{u} \times \boldsymbol{B}$ fields would be several kilovolts (kV) for a typical Sq vortex. We can get to this estimate in a similar manner as the estimate of the 0.5–2 GW Joule heating by $Sq$ was done in the manuscript: For a total Hall current $I_H$ of $\approx 100$ kAa $\langle \Sigma_H \rangle \approx 12$ S the potential drop $I_H/\langle \Sigma_H \rangle$ across the $Sq$ vortex becomes about 8 kV. Without the compensating $E^*$ the potential difference between hemispheres would reach up to twice this value, about 16 kV! The plasma sets up the electrostatic $E^*$

[Figure]

**Figure 1.** 2-d view like in Figure 5, in the reference of the southern neutral gas. The electric fields are such that for asymmetric conductances $\Sigma_N = 0.5$ and $\Sigma_S = 1.0$ the same current $\boldsymbol{J}$ is obtained. $\boldsymbol{J} \times \boldsymbol{B}$ force and magnetic stress $\Delta\boldsymbol{B}$ are omitted in this Figure. The sizes of the symbols for Poynting flux in Figure 5 and this Figure are according to the flux magnitudes having the same scale.

and so short-circuits potential drops along B. This causes FACs. This means that small remaining voltage drops along B of the order of Volts cannot be avoided. The magnitude of these remaining voltages were estimated already by Fukushima (1979).

I thank again Prof. Fujii for the discussion on this subject.

5    – the analogy with quantum mechanics is perhaps overemphasized, particularly the reference Greenberger, Horne, and Zeilinger (1989) is only little relevant and can be removed. The analogy is meant purely linguistically in the sense that a system having two components is described using the adjective "entangled." My intention is to distinguish from "coupled" dynamos, which would exist independently, but interact because they happen to be on the same field-line. The described dynamos cannot exist independently without any "entanglement".

10    But for future work the "entanglement" concept can be generalized to three and more dynamo layers in the ionosphere.

---

## Referee Comment (RC1) · Octav Marghitu (Referee) · 21 Nov 2019

The paper "Entangled dynamos and Joule heating in the Earth's ionosphere" provides a model of energy generation and dissipation at mid latitudes, based on interhemispheric asymmetry of neutral winds and on the assumption that magnetic field lines are, essentially, equipotential. A key feature of the model is the use of two reference systems, moving separately with the neutral atmosphere at the two conjugate ends of the magnetic field line. It is pointed out that these two systems are the right choice to investigate energy dissipation, complemented by energy generation in the opposite hemisphere, while any other aether-like system, e.g., a system fixed to the Earth, is arbitrary and not appropriate to address the matter.

[Figure]

I fully agree with this perspective and, altogether, I see the paper as a valuable contribution to the field. I shall be happy to recommend publication, as soon as the issues below are taken care of.

1. Using two reference systems has obvious merits, but it is at the same time challenging, in particular by introducing two instances of the Poynting flux. This is detailed in the Discussion section, though I think that the clarity of the message may benefit from sub-sectioning and some re-arrangement:

1a. More specifically, the three paras from p.12, L17 ("We claim that...") up to p.13, L21 ("...and heating effects") could be moved to p.9, L 26, after the para describing the main features of the model. The first part of Section 5, up to this point, together with the three paras, could make the first sub-section of the Discussion, emphasizing the need for two reference systems.

1b. The rest of the Discussion could be organized in two more sub-sections, one on quantitative estimates of Sq Joule heating (from p.9, L26 up to p.12, L16), and one on applying the model to high latitudes (from p.13, L22 to the end of the Section).

2. Speaking about high latitudes, these are associated in the paper with open field lines, both in the last part of the Discussion and in the Conclusions (e.g., p. 16, L8). As a matter of fact, much of the energy dissipation takes place in the auroral region, which is believed to be threaded (mostly) by closed field lines, that connect the two hemispheres via the plasma sheet in the magnetosphere. However, in this case plasma parameters do not preclude any more parallel electric fields (e.g., much lower density compared to plasmasphere). The open field lines are in general associated with the polar cap, where energy dissipation is limited. As of now, the discussion on high latitudes refers mainly to open field lines / polar cap, while the specific case of the auroral region is just touched a bit, implicitly, in the second last para of the Conclusions. Please complete the Discussion and Conclusions by addressing explicitly the auroral region, where the key feature is the parallel electric field on closed field lines.

[Figure]

3. The mapping between the two hemispheres could be emphasized by adding the two respective reference systems, (x, y, z), N and S, on the side of Figs 3 and 4, with the x axis pointing northward in N and southward in S. This would also clarify the '+' sign in Equation 5. It would help as well to add J_N and J_S explicitly before Eq. (5), J_N=\Sigma_N EˆstarstarN and J_S=-\Sigma_S Eˆstar_S.

4. The proxy in Eq. (15) is probably derived by assuming that ion-neutral collision frequency and ion gyro-frequency are roughly equal in the dissipation layer \Delta z. Please make this clear.

5. I presume the final text will include also the adjustments added in the Author comment – which are not addressed any more here.

6. Minors

p.2, L12: interferE

L15: conductance => conductivity

p.3, Fig. 1: Please increase the figure (zonal wind arrows are not visible) and font size (in particular for the Legend).

L4: Perhaps complete the sentence with: ". . . on the E side, which is the standard form of Lorentz transformation for non-relativistic velocities, u."

Eq. (3): Delete \Sigma_P in the second term.

p.4, L13: . . .connecting either the latitude lines '1', or the latitude lines '2', or both.

L15: are from of Galilei

p.5, L9: current => FACs

p.6, L3: frame => frame with

L6: surrendered => relaxed (?)

L11: an opportunity => a stronger motivation (?)

p.7, L1: closeS

p.9, L27: and or or

L29: and simulations => nor simulations (?)

p.10, L9: but => therefore

L15: from a with

L22: given => given as well (?)

L32: Delete 'also'. Please explain briefly 'opposite polarity'.

p.11, L30: integrated => integrated over

p.12, L23: to the load => to the load in the opposite hemisphere

L30: as a being

p.13, Eq. (16): According to Eq. (2), I think this should be written as Eˆstar(z)-u(z)B(z)=const. (if mapping is neglected), i.e., electric field in a given, unique reference system, is constant.

L19: the describe . . . confirm Figure 1

L27: It is well accepted => please provide reference.

p.14, Eq. (17): The '+' sign on the r.h.s. should be '-', similar to Eq. (2).

Eq. (18): Both '+'signs on the r.h.s. should be '-': the first, same as above; the second, satellite velocity with respect to neutral atmosphere is v_orb - u.

L23: Considered => Considered first (?)

---

## Referee Comment (RC2) · Anonymous Referee #2 · 13 Jan 2020

Review comments on the manuscript MS angeo-2019-71: "Entangled Dynamos and Joule Heating in the Earth's Ionosphere" by S. Buchert

The author has provided a detailed description of current generation and dissipation processes in the mid-latitude ionosphere. An important message is the fact that a correct picture only emerges when the processes at conjugate locations in the hemispheres are considered simultaneously. Even though this is a valid claim, the way it is presented is not intuitive and obviously difficult to digest for readers. An indication for that is the long time (about 8 months) and many rejections it took to find referees for the paper. It definitely needs improvement. Otherwise one may ask, what is the purpose of a paper that nobody understands and not taken notice of.

General comments

[Figure]

The presentation of examples could be a little bit more constructive and easier under-standable of the readers. It is good that scenarios in different reference frames are outlined, but it would be helpful to focus more on the frame independent quantities. These are, e.g. B-fields, currents, energy dissipation, and velocity difference between plasma drift and wind velocity. I find it not helpful when you state that in the case of Fig. 3 the NH is the sink and SH the dynamo and in case of Fig.4, where you just have changed the reference frame, NH is the dynamo and SH the dynamo. You should have described what actually happens, it is the competing wind-generated E-fields in the two hemispheres that prevents the plasma from moving thus gives equal frictional heating in both hemispheres.

In the case of different conductances (Figs. 5 and 6) you correctly state that power dissipation is higher in the low conductivity hemisphere. Both these examples had been much easier to be understood if you had added also the plasma drift and calculated the frictional heating in the hemispheres.

Specific comments

Abstract: The dynamo effect is not limited to different winds in the two hemispheres. Also differences in conductivity, B-field strength, field configuration, etc. can be respon-sible generating currents.

Pg. 9, line 7: In the past versions of first-principle ionospheric electrodynamic mod-els the relation $E + u \times B = 0$ was actually maintained by adjusting the wind ve-locity u. In the latest version of TIEGCM also other currents such as gravity-driven or plasma pressure gradient currents are considered. Therefore, these models now have a 3D electrodynamic solver that maintains current continuity and equal po-tentials at conjugate locations. For more details see Richmond and Maute (2014) doi:10.1002/9781118704417.ch6

Line 25: I would suggest to change to "...dependence only on relative motion between plasma and neutral gas, no reference to absolute frames."

Lines 26-29: It is not clear to me what these sentences want to state.

Pg. 10, lines 24-28: Here again, it would be instructive to address also the difference between plasma drift and wind. In particular, since the local plasma drift is the prime measurement of satellites in the ionosphere, not E-field.

Pg. 11, line 5: Concerning inter-hemispheric field-aligned currents (IHFAC) there are more recent results of their mean properties, e.g. Lühr et al. (2020) doi:10.1002/2019JA027419. Furthermore, it has been noticed that these IHFACs do not originate from the Sq focus but there is a group of IHFACs located equatorward of the focus, and another group of IHFACs with mainly opposite current directions is emanating from mid-latitudes above the focus (see Park et al., 2020, accepted, doi:10.1002/2019JA027694)

Line 10: The sentence correctly states that wind energy is extracted from one hemisphere and dissipated as Joule heating in the other. But unfortunately, no estimate of the energy transfer from the summer to the winter hemisphere, relative to the total energy, is given. Only the total energy is estimated. Here again we like to stress the very different IHFAC configurations for June and December solstices although no such seasonal differences are obvious from ground-based maps of Sq patterns.

Pg. 12, lines 19ff: You start again stressing the frame dependence of Poynting flux. This is for me the wrong definition. Poynting flux as such is frame independent. Here again the velocity differences between plasma and neutral in both hemispheres would give a unique picture.

Pg. 15, line 2: It is not clear what is meant by "an isolated neutral wind in a plasma would not result in any steady state dynamo effect."

Lines 4-9: I cannot agree with the suggested principle of Sq generation. The mid-latitude winds are only marginally affected by the plasma dynamics. Therefore, it is the difference in plasma drift response to the winds in conjugate points (depending

on conductivity, B-field strength, wind velocity, etc.) that is communicated along field lines between the hemispheres. Again, the resulting local velocity difference between plasma and neutrals drives the electrodynamic processes. The 12-hour period of the Sq signal is mainly dictated by the atmospheric semidiurnal tide, which is clearly dominating at mid latitudes. Longitudinal variations of the various involved quantities play only a minor role.

Last line: As mentioned above, the 3D electrodynamic solver in TIEGCM avoids potential drops between conjugate points.

---

## Author Comment (AC2) · 20 Feb 2020

**Reply to Octav Marghitu's comment on "Entangled Dynamos and Joule Heating in the Earth's Ionosphere" by**

Stephan C. Buchert[1]

[1]Swedish Institute of Space Physics, Uppsala, Sweden

**Correspondence:** Stephan Buchert (scb@irfu.se)

**1 Replies to comments by Referee 1**

Cited referee comments are in red, replies in magenta.

I thank the referee, Octav Marghitum, for the interest in the manuscript and the time spent reading it, and for the helpful comments.

5    1. Using two reference systems has obvious merits, but it is at the same time challenging, in particular by introducing two instances of the Poynting flux. This is detailed in the Discussion section, though I think that the clarity of the message may benefit from sub-sectioning and some re-arrangement:

1a. More specifically, the three paras from p.12, L17 ("We claim that. . .") up to p.13, L21 (". . .and heating effects") could be moved to p.9, L 26, after the para describing the main features of the model. The first part of Section 5,

10   up to this point, together with the three paras, could make the first sub-section of the Discussion, emphasizing the need for two reference systems.

I have moved the discussion on the Poynting flux in different references frames to the location suggested by the referee. Also it is slightly modfied:

We claim that there is Poynting flux from $N$ to $S$ as well as from $S$ to $N$, each transporting electrodynamic energy

15   from a dynamo to a load. Adding both Poynting fluxes would give zero (in the symmetric case), but this is not a meaningful view. The Poynting flux $\boldsymbol{S} = \boldsymbol{E} \times \Delta \boldsymbol{B} / \mu_0$, where $\boldsymbol{E}$ includes the motional field, is frame dependent, as well as the term $\boldsymbol{J} \cdot \boldsymbol{E}$. There are infinitely many possible reference frames, and in each of these Poynting's theorem is of course valid. But only frames with the physical material at rest, in this case of zero neutral wind are special, are the "laboratory frame" with the $\boldsymbol{J} \cdot \boldsymbol{E}^*$ term and the ionospheric Ohm's law giving the dissipation. We argue that it

20   is in this frame where $\boldsymbol{J} \cdot \boldsymbol{E}$ represents the neutral dynamo's power in $\mathrm{Wm}^{-2}$ and the Poynting flux the amount and direction of electromagnetic energy being transported from the dynamo to the load. On each magnetic flux tube the neutral winds at each conjugate end define so two "laboratory" frames connected to physical material. In each of the two frames one end is the location of the load. At the other end is a dynamo where $\boldsymbol{J} \cdot \boldsymbol{E} = \boldsymbol{J} \cdot (\boldsymbol{E}^* - \Delta \boldsymbol{u} \times \boldsymbol{B}) < 0$ matching the dissipation at the load. When switching the reference frames the roles also switch, and the Poynting

flux between both ends flips to the opposite direction. The neutral dynamo power is so determined by the neutral wind difference at the conjugate points.

1b. The rest of the Discussion could be organized in two more sub-sections, one on quantitative estimates of Sq Joule heating (from p.9, L26 up to p.12, L16), and one on applying the model to high latitudes (from p.13, L22 to the end of the Section).

Subjections were added to the long section "Discussion":

**1.1   The Model of Entangled Dynamos**

**1.2   Estimation of the Joule Heating Power**

**1.3   The Atmosphere, a Dynamo for Space?**

2. Speaking about high latitudes, these are associated in the paper with open field lines, both in the last part of the Discussion and in the Conclusions (e.g., p. 16, L8). As a matter of fact, much of the energy dissipation takes place in the auroral region, which is believed to be threaded (mostly) by closed field lines, that connect the two hemispheres via the plasma sheet in the magnetosphere. However, in this case plasma parameters do not preclude any more parallel electric fields (e.g., much lower density compared to plasmasphere). The open field lines are in general associated with the polar cap, where energy dissipation is limited. As of now, the discussion on high latitudes refers mainly to open field lines / polar cap, while the specific case of the auroral region is just touched a bit, implicitly, in the second last para of the Conclusions. Please complete the Discussion and Conclusions by addressing explicitly the auroral region, where the key feature is the parallel electric field on closed field lines.

The manuscript clarifies how the atmosphere dynamo works and what its effects are. It is not intended as new comprehensive theory/model of interaction between ionosphere-thermosphere and space, and of auroral processes. In the latter the atmosphere dynamo is not expected to play an important role. To isolate the atmosphere dynamo from ionosphere-magnetosphere coupling, the plasma, including the one all along closed field-lines between conjugate points, is assumed to be "passive", i. e. it only reacts to neutral dynamics. Observationally the Sq perturbations are clearly visible on quiet days even up to high latitudes, suggesting that the isolated treatment of Sq is in principle testable. On moderate to active days Sq gets buried in larger geomagnetic disturbances even at mid-latitudes. Then the space plasma is not "passive" but has its own dynamics including, at times, the parallel fields mentioned by the referee. The high latitudes are mentioned in the manuscript, because it had been suggested that particularly there, probably on open field-lines, the atmosphere dynamo would transport energy into space, statistically, on average. Also this would be a small effect having probably little relation with aurora and parallel electric fields. The co-existence of Sq driven by entangled atmosphere dynamos with substorms, auroras etc is acknowledged in the $2^{nd}$ last paragraph which is slightly reformulated:

On closed field-lines the currents and fields of entangled dynamos can coexist with currents and fields induced by plasma motion in the magnetosphere driven by interaction with the solar wind, to use a generic term. This

includes sub-storms, including auroral features sometimes associated with E-fields parallel to $\boldsymbol{B}$, high-latitude plasma convection, its occasional penetration towards lower latitudes etc.

3. The mapping between the two hemispheres could be emphasized by adding the two respective reference systems, (x, y, z), N and S, on the side of Figs 3 and 4, with the x axis pointing northward in N and southward in S. This would also clarify the '+' sign in Equation 5. It would help as well to add $J_N$ and $J_S$ explicitly before Eq. (5), $J_N = \Sigma_N E_N^*$ and $J_S = -\Sigma_S E_S^*$.

Labeled arrows for the $X$ and $Y$ axis were added to Figures 5–6. Equation 5 was added as the referee suggests:

. . . for the current calculation the frames in $N$ and $S$ are not the same:

$$J_N = \Sigma_N E_N^*, \ J_S = -\Sigma_S E_S^*; \tag{5}$$

$$J_N + J_S = \Sigma_N E_N^* + \Sigma_S E_S^* = 0 \tag{6}$$

4. The proxy in Eq. (15) is probably derived by assuming that ion-neutral collision frequency and ion gyro-frequency are roughly equal in the dissipation layer $\Delta z$. Please make this clear.

Text is added:

. . . with $B = 35000$ nT as an average value of the magnetic field strength at mid latitudes and the factor $e/2B$ giving the conductivity where ion gyro and ion-neutral collision frequencies are equal.

p.3, Fig. 1: Please increase the figure (zonal wind arrows are not visible) and font size (in particular for the Legend).

I have revised the Figure. Now it shows the neutral wind relative to the Earths continents, as we tend to imagine it. Later in the manuscript it is argued that the Earth fixed system is actually irrelevant, only wind differences matter.

L4: Perhaps complete the sentence with: ". . . on the E side, which is the standard form of Lorentz transformation for non-relativistic velocities, u."

The sentence is completed:

Please note that in many publications this equation is written with the $+\boldsymbol{u} \times \boldsymbol{B}$ term on the $\boldsymbol{E}$ side , which is the standard form of the Lorentz transformation for non-relativistic velocities $\boldsymbol{u}$.

Eq. (3): Delete $\Sigma_P$ in the second term.

The second term should not have a factor $\Sigma_P$, it is deleted.

p.4, L13: . . . connecting either the latitude lines '1', or the latitude lines '2', or both.

The statement is modfied to:

But this configuration of $\boldsymbol{E}$ implies a potential drop along magnetic field lines connecting either latitude circles "1" or latitude circles "2" or along both these field lines.

L15: are from of Galilei

changed to:

We  reject the initial idea that the only electric fields are those of Galilei coordinate transformations from neutral to observer frames.

p.5, L9: current => FACs

Changed.

p.6, L3: frame => frame with

"with" is added.

L6: surrendered => relaxed (?) changed.

L11: an opportunity => a stronger motivation (?) changed.

p.7, L1: closeS: closes

p.9, L27: and or or

The statement is deleted after comment by referee 2. Instead text in section "Conclusions and Outlook", $9^{th}$ paragraph outlines how a computer algorithm could handle relative neutral wind differences in a way that is consistent with the theory described in the manuscript:

L29: and simulations => nor simulations (?)

The statement is deleted after comment by referee 2.

p.10, L9: but => therefore changed.

L15: from a with

The statement is changed to:

. . . and deviations from a field that is with respect to the dipole equator perfectly symmetric .

L22: given => given as well (?)

The statement is changed to:

Other explanations for the semi-diurnal component in Sq have been given as well (confirm Yamazaki and Maute, 2017).

L32: Delete 'also'. Please explain briefly 'opposite polarity'.

"polarity" is changed to "direction", "also" deleted

p.11, L30: integrated => integrated over "over" added.

p.12, L23: to the load => to the load in the opposite hemisphere

"in the opposite hemisphere" is added.

L30: as a being "a" is deleted.

p.13, Eq. (16): According to Eq. (2), I think this should be written as E^star(z)- u(z)B(z)=const. (if mapping is neglected), i.e., electric field in a given, unique reference system, is constant.

Correct, the sign is changed. I think that this is the mapping condition, at least for the case of only zonal winds.

L19: the describe . . . confirm Figure 1

"prescribed" is changed to "described".

L27: It is well accepted => please provide reference.

"It is well accepted" is deleted. Perhaps surprisingly I could not find a reference where this is explicitly stated, and also referee 2 had objections.

p.14, Eq. (17): The '+' sign on the r.h.s. should be '-', similar to Eq. (2).

Eq. (18): Both '+'signs on the r.h.s. should be '-': the first, same as above; the second, satellite velocity with respect to neutral atmosphere is $v_{o}rb - u$.

All signs are changed, I had myself become confused by the different notation than Kelley's.

L23: Considered => Considered first (?)

Throughout the manuscript only a "passive" plasma is considered, adding "first" would not fit.

**2 Other Changes**

Changes were made according to comments by referee 2, please confirm the reply for a list.

According to my own comment in section "Preliminaries"

.

was deleted. Fukushima's contribution is reformulated as:

Fukushima (1979) had suggested that there are electric potential differences between conjugate points of only a few Volts.

References that were added are:

Cosgrove, R. B., Bahcivan, H., Chen, S., Strangeway, R. J., Ortega, J., Alhassan, M., Xu, Y., Welie, M. V., Rehberger, J., Musielak, S., and Cahill, N.: Empirical model of Poynting flux derived from FAST data and a cusp signature, Journal of Geophysical Research: Space Physics, 119, 411–430, https://doi.org/10.1002/2013JA019105, 2014.

Drob, D. P., e. a.: An update to the Horizontal Wind Model (HWM): The quiet time thermosphere, Earth and Space Science, 2, 301–319, https://doi.org/10.1002/2014EA000089, 2015.

Richmond, A. D.: On the ionospheric application of Poynting's theorem, Journal of Geophysical Research: Space Physics, 115, https://doi.org/10.1029/2010JA015768, 2010.

---

## Author Comment (AC3) · 20 Feb 2020

**Reply to anonymous Referee 2's comment on "Entangled Dynamos and Joule Heating in the Earth's Ionosphere" by**

Stephan C. Buchert[1]

[1]Swedish Institute of Space Physics, Uppsala, Sweden

**Correspondence:** Stephan Buchert (scb@irfu.se)

**1  Replies to comments by Referee 2**

Cited referee comments are in red, replies in magenta.

I thank the referee for the interest in the manuscript and the time spent reading it, and for the helpful comments. The referee's main objection is that the manuscript is difficult to understand. She or he then summarizes "that a correct picture only emerges when the processes at conjugate locations in the hemispheres are considered simultaneously." Yes, this is the main point regarding the Sq system, and I'm relieved that at least this point has been reasonably comprehensible.

The presentation of examples could be a little bit more constructive and easier understandable of the readers. It is good that scenarios in different reference frames are outlined, but it would be helpful to focus more on the frame independent quantities. These are, e.g. B-fields, currents, energy dissipation, and velocity difference between plasma drift and wind velocity. I find it not helpful when you state that in the case of Fig. 3 the NH is the sink and SH the dynamo and in case of Fig.4, where you just have changed the reference frame, NH is the dynamo and SH the dynamo. You should have described what actually happens, it is the competing wind-generated E-fields in the two hemispheres that prevents the plasma from moving thus gives equal frictional heating in both hemispheres.

**Reply**

One of the really important points is that a frame-independent electrostatic $E^*$ is created as a result of neutral wind differences at conjugate points. So $E^*$ is one of the frame independent quantities and it is THE focus of the manuscript.

To make it clearer, that I'm moving away from the frame-dependent quantities and focusing on frame independence, Undoubtedly Sq variations have to do with neutral motion, but a neutral wind $\boldsymbol{u}$ and associated motional field $\boldsymbol{u} \times \boldsymbol{B}$ is frame dependent. In the frame of the neutral gas both are zero. So what exactly drives the Sq currents and fields?

is added in section Introduction, and

It is here important to note that $\boldsymbol{E}^*$ is a frame-independent electrostatic field driving currents according to Equation 1. The frame-dependent motional $\boldsymbol{u} \times \boldsymbol{B}$ does not drive any currents, it is not a real field.

is added to section "Preliminaries".

The referee points out the significance of the velocity difference between plasma drift and wind velocity. No doubt that this velocity difference is important. However, its magnitude and direction are a complicated functions of the ratio of the ion-neutral collision and the ion gyro frequencies. At the bottom of the dynamo region the velocity difference vector is small and in the direction of the frame independent $\boldsymbol{E}^*$, and at the top it is $\boldsymbol{E}^* \times \boldsymbol{B}/B^2$, with a transition in both magnitude and flow angle in between. To describe mathematically the velocity difference between plasma drift and wind velocity would require to discuss this issue and involve equations that are much more complicated than the ones given for the $E^*$, Equations 4–7. To include the velocity difference between plasma drift and wind velocity in the Figures 3–6. I would need to show the vectors for a specific ratio of these frequencies and decide which ratio. These complications which would not contribute to a better understanding are circumvented by the commonly well-know height integration (confirm section "Preliminaries", $1^{st}$ paragraph). In other words, the ionosphere in each hemisphere is treated as "thin". Moreover, adopting the $E$ and $j$ paradigm (confirm section "Preliminaries", $2^n d$ paragraph) this velocity difference between plasma drift and wind velocity is only an effect of $E^*$. The cause of everything is non-mapping neutral gas velocities at conjugate points. Therefore the velocity difference between plasma drift and wind velocity is admitted:

Instead, the plasma will establish an electric field $\boldsymbol{E^*}$ (perpendicular to $\boldsymbol{B}$)  such that potentials along $\boldsymbol{B}$ are avoided. The non-zero $\boldsymbol{E^*}$ implies that the plasma in the plasmasphere drifts, and that there is a velocity difference between plasma and neutral gas. We  reject the initial idea that the only electric fields are from of Galilei coordinate transformations from neutral to observer frames.

The velocity difference between plasma drift and wind velocity is not quantified further in the equations and shown in the Figures, for the reasons described above.

Abstract: The dynamo effect is not limited to different winds in the two hemispheres. Also differences in conductivity, B-field strength, field configuration, etc. can be responsible generating currents.

**Reply:**

Simple wind differences in the two hemispheres are the exclusive driver indeed only for a symmetric magnetic field, like a centered dipole (which is assumed for the examples and equations in the manuscript). If an asymmetric B-field is considered, like a non-centered dipole, then, for the 1-d case of only zonal winds, instead of a wind difference $u_N - u_S$ the expression $u_N B_N - u_S B_S$ has to be non-zero in order to drive a dynamo (rather entangled dynamos). Differences in conductivities are discussed in the section "Asymmetric Dynamos".

I insist that differences in conductivities do not generate the currents, and they are exclusively caused by relative wind differences at conjugate points. Differences in conductivities only affect the magnitude of non-zero currents, how strong the Joule heating is, and how it is partitioned between the hemispheres. A dynamo effect is limited to neutral winds that do not map at conjugate points. Differences in the B-field strength and configuration at conjugate points

affect the mapping condition. For a symmetric B-field the mapping conditions is simply that the wind difference is zero. There is no dynamo effect if winds are (for symmetric B-field exactly) mirror-symmetric between magnetic hemispheres. Such non-dynamo winds may have complicated structures like vortices, shears etc., still they don't have any dynamo effect or cause magnetic perturbations.

5    The abstract is modified:

. . . where a dynamo effect is obtained only in case of winds perpendicular to the magnetic field $\boldsymbol{B}$ that  do not map along $\boldsymbol{B}$.  Winds where $\boldsymbol{u} \times \boldsymbol{B}$ is constant have no effect.

In section "Asymmetric Dynamos" (page 9, lines 1–5) I appended

Asymmetry can also be in the magnetic field, with different field strengths in both hemispheres, $B_N \neq B_S$. Rather
10    than the simple difference $\Delta u$ then winds at conjugate points don't map if

$$\Delta w = u_{y,N} B_N - u_{y,S} B_S$$

is not zero, and $\Delta w$ replaces $\Delta uB$ in Equations 4–11. A magnetic asymmetry between hemispheres changes the mapping condition, but it does not cause asymmetry of $E^*$ or Joule heating.

Pg. 9, line 7: In the past versions of first-principle ionospheric electrodynamic models the relation E+uxB = 0
15    was actually maintained by adjusting the wind velocity u. In the latest version of TIEGCM also other currents such as gravity-driven or plasma pressure gradient currents are considered. Therefore, these models now have a 3D electrodynamic solver that maintains current continuity and equal potentials at conjugate locations. For more details see Richmond and Maute (2014) doi:10.1002/9781118704417.ch6

**Reply:** I have deleted

20

Instead text in section "Conclusions and Outlook", $9^{th}$ paragraph outlines how a CGM computer algorithm could handle relative neutral wind differences in a way that is consistent with the theory described in the manuscript:

A numerical simulation that applied directly the motional field $\boldsymbol{u} \times \boldsymbol{B}$ to calculate currents would be incorrect.
25    Instead the relative neutral winds (and $\boldsymbol{B}$) at both conjugate points should and can be used to obtain the frame-independent $\boldsymbol{E}^*$, Equations 6–7 for the here discussed very simplified case of no meridional winds and symmetric $\boldsymbol{B}$. $\boldsymbol{E}^*$ drives the current according to Ohm's law, Equation 1. For purely zonal neutral winds and symmetric $\boldsymbol{B}$ Equations 6–8 apply.

Line 25: I would suggest to change to ". . .dependence only on relative motion between plasma and neutral gas,
30    no reference to absolute frames."

**Reply:** The text is changed to

1. . . .

2. and dependence only on relative  differences of the neutral wind $\Delta u$ and between plasma and neutral gas, no reference to  an absolute neutral wind $u$.

The point in the manuscript is that a relative motion of the neutral gas at conjugate points, i. e. $u_N \neq u_S$, is the cause. It induces a relative motion between neutrals and plasma, currents and Joule heating. So relative motion between plasma and neutral gas is only an effect, not the cause.

Lines 26–29: It is not clear to me what these sentences want to state.

**Reply:** I have deleted the original lines 26–29, please see the text on numerical simulations added in section "Conclusions and Outlook", $9^{th}$ expressing in a better way what I wanted to state.

Pg. 10, lines 24-28: Here again, it would be instructive to address also the difference between plasma drift and wind. In particular, since the local plasma drift is the prime measurement of satellites in the ionosphere, not E-field.

**Reply:** The text has been changed to

However, an electrostatic $\boldsymbol{E}^*$ and corresponding relative motion between $u$ and plasma must exist to drive the interdynamo currents (equation 8 as well as any Hall currents. A non-zero $\boldsymbol{E}^*$ is not created by a local non-zero $u$ in the Earth-fixed frame. It has a non-local origin, for example  when the local thermospheric wind is zero relative to the observatory, but strong at the conjugate point. No effect is observed, if there is a strong local thermospheric wind, and the same strong wind at the conjugate point.

The difference between plasma drift and wind is now mentioned, but not included in Equations and Figures, for the reasons described above. The text is intended to highlight the non-local cause of the Sq variations. Again, I oppose to the notion that the dynamo is because of local neutral wind in some absolute reference frame. Rather it is caused by differences in the neutral motion along a magnetic field line. This brings in a non-local origin/cause of the relative plasma-neutral drift and $E^*$. Point measurements of drift or E-field with a single satellite exist of course, but they do not directly reveal this non-locality, and I need to argue in such a theoretical manner which is perhaps difficult to understand.

Pg. 11, line 5: Concerning inter-hemispheric field-aligned currents (IHFAC) there are more recent results of their mean properties, e.g. Lühr et al. (2020) doi:10.1002/2019JA027419. Furthermore, it has been noticed that these IHFACs do not originate from the Sq focus but there is a group of IHFACs located equatorward of the focus, and another group of IHFACs with mainly opposite current directions is emanating from mid-latitudes above the focus (see Park et al., 2020, accepted, doi:10.1002/2019JA027694)

**Reply:** I have added both references. The model of a jet-like zonal wind difference between conjugate points does show the IHFAC at the edges of the neutral wind jets (pse see Figures 3–6). This model is constructed to show the dynamo principle in the simplest possible configuration. It is not meant to have all the important elements of the real Sq. But if I imagine the interhemispheric neutral wind difference as two large vortices, one in each hemisphere, then FACs should connect the edges of the vortices. The inner edges would be circular-like around the focui. A polar orbiting satellite should then detect between equator and pole at mid-laitudes two pairs of FACs with opposite polarity. This seems to me similar as Park and Lühr (2020) describe their results. By closing the jet-like $\Delta u$ from the manuscript into a vortex the results might become more consistent with the latest Swarm analysis. However,

treating such a more complicated, albeit more realistic configuration is beyond the scope of the manuscript where only sketch-like, analytical solutions are presented.

Line 10: The sentence correctly states that wind energy is extracted from one hemisphere and dissipated as Joule heating in the other. But unfortunately, no estimate of the energy transfer from the summer to the winter hemisphere, relative to the total energy, is given. Only the total energy is estimated. Here again we like to stress the very different IHFAC configurations for June and December solstices although no such seasonal differences are obvious from ground-based maps of Sq patterns.

**Reply:** The Joule heating in each hemisphere is given in equations 9 and 10. Depending on assumptions for the seasonal variations of the Pederen conductances the seasonally varying energy transfer could be estimated. However, I think, that a more elaborate modeling would be needed to get results that could be meaningfully compared with the Swarm results. This would be out of the scope of this manuscript. A non-aligned and also non-centered dipole axis should result in differences in the IHFACs between June and December solstices, but I cannot say how large the effect would be. Ground-based Sq maps and IHFACs measured in LEO would differ if the ratio between Hall and Pedersen conductances is not constant and depends on season and $\boldsymbol{B}$. This is certainly the case, but again it is difficult to assess how large the effect would be without further more detailed investigation.

Pg. 12, lines 19ff: You start again stressing the frame dependence of Poynting flux. This is for me the wrong definition. Poynting flux as such is frame independent. Here again the velocity differences between plasma and neutral in both hemispheres would give a unique picture.

**Reply:**

The Poynting flux $\boldsymbol{S}$ is defined as $\boldsymbol{E} \times \boldsymbol{B}/\mu_0$, with $\boldsymbol{E}$ including the motional field, as it would be measured by an instrument resting in this frame. Thus $\boldsymbol{S}$ is frame dependent. I'm not aware that other definitions have been suggested or used anywhere in the literature, and why the definition used in the manuscript should be "wrong".

An alternative definition would always use the frame-independent field: $\boldsymbol{S}^* = \boldsymbol{E}^* \times \boldsymbol{B}$. $\boldsymbol{S} = \boldsymbol{S}^*$ only in the frame of the neutral gas. Then Poynting's theorem (for the stationary case) $\nabla \cdot \boldsymbol{S}^* = -\boldsymbol{J} \cdot \boldsymbol{E}^*$ always describes an energy transfer into the ionosphere and dissipation by Joule heating. As mentioned in section "Introduction", $\boldsymbol{J} \cdot \boldsymbol{E}^* >= 0$ always according to Ohm's law. Then Poynting's theorem would not allow for a dynamo where $\boldsymbol{J} \cdot \boldsymbol{E} < 0$. This doesn't seem right to me.

The velocity differences between plasma and neutral in both hemispheres are of course unique and frame-independent, but they describe always friction, which is another name for Joule heating (Vasyliunas and Song, 2005). Only with the frame dependent definition of $\boldsymbol{S}$, as in the manuscript, the complete picture with dynamos transfering the generated energy to the loads into the opposite hemispheres becomes clear.

Pg. 15, line 2: It is not clear what is meant by "an isolated neutral wind in a plasma would not result in any steady state dynamo effect."

**Reply:** The text has been deleted. Instead the first paragraph of section "Conclusions and Outlook" states what I think is the overall picture:

It is not the neutral wind itself, defined as any non-zero neutral velocity $\boldsymbol{u}$ in an Earth-fixed frame that causes a dynamo. Rather relative neutral gas motions which do not map between magnetically conjugate points drive Sq currents, magnetic pertubations and Joule heating. A wind system that is mirror symmetric across the magnetic equator, for symmetric $\boldsymbol{B}$, does not act as dynamo. Lorentz forces $\boldsymbol{j} \times \boldsymbol{B}$ drive the wind system towards such symmetry while the solar heat input and non-inertial (Coriolis) forces not aligned to the geomagnetic field drive it away.

Lines 4-9: I cannot agree with the suggested principle of Sq generation. The midlatitude winds are only marginally affected by the plasma dynamics. Therefore, it is the difference in plasma drift response to the winds in conjugate points (depending on conductivity, B-field strength, wind velocity, etc.) that is communicated along field lines between the hemispheres. Again, the resulting local velocity difference between plasma and neutrals drives the electrodynamic processes. The 12-hour period of the Sq signal is mainly dictated by the atmospheric semidiurnal tide, which is clearly dominating at mid latitudes. Longitudinal variations of the various involved quantities play only a minor role.

**Reply:** I have clarified in the replies and changes to the manuscript that the wind differences are the ultimate cause of, and the referee can hopefully agree to this suggested principle of Sq generation. The dependence on conductances and B-field strength pointed out by the referee is discussed in the manuscript, also that these are not the cause, not a ncessary condition. That the local velocity difference between plasma and neutrals is a result of the winds is nowhere disputed in the manuscript. The velocity difference indicates friction and generation of heat which is the same process as what is commonly also called Joule heating, the name used in the manuscript. Lines 4-9 in the original manuscript do not describe the suggested principle of Sq generation, this is described before. The lines point out an anticipated consequence from the peculiar misalignment of the geomagnetic field with respect to the rotation axis, namely a 12 hour modulation. The misalignment is not necessary for Sq currents, it only adds an expected semidiurnal component. For example, Saturn has no axial misalignment but IHFACs were detected and attributed to wind differences at conjugate points (Khurana et al., 2018).

The lines are revised to:

We suggest that the Earth's magnetic Sq variations are driven by neutral wind differences at conjugate points. The main dipole geomagnetic field is tilted with respect to the Earth's rotation axis as well as it is not centered, making it a strongly misaligned rotator. This  might contribute to the presence of a 12-hour component in Sq variations. Drob et al. (2015) state that the average neutral wind is partially, mostly at high latitudes, magnetically aligned even at quiet time. $\boldsymbol{J} \times \boldsymbol{B}$ forces of the entangled dynamos, confirm Figures 5–6 act to align to neutral wind to magnetic coordinates, while pressure gradients caused by solar EUV and Coriolis forces have no geomagnetic relation. The dynamo currents are modulated by the product of the Pedersen conductances in both hemispheres resulting also in a 24 hour component of the variations at a fixed point on the Earth. In addition the Sq variations reflect of course also dynamics of the neutral atmosphere itself including any semidiurnal component .

Last line: As mentioned above, the 3D electrodynamic solver in TIEGCM avoids potential drops between conjugate points.

**Reply:** As described alreay in replies above the text has been replaced to describe how numerical calculations would need to be done in order to be consistent with manuscript and physically correct. The text does not refer specifically to TIEGCM.

**2    References**

Drob, D. P., e. a.: An update to the Horizontal Wind Model (HWM): The quiet time thermosphere, Earth and Space Science, 2, 301–319, https://doi.org/10.1002/2014EA000089, 2015.

Khurana, K. K., Dougherty, M. K., Provan, G., Hunt, G. J., Kivelson, M. G., Cowley, S. W. H., Southwood, D. J., and Russell, C. T.: Discovery of Atmospheric-Wind-Driven Electric Currents in Saturn's Magnetosphere in the Gap Between Saturn and its Rings, Geophysical Research Letters, 45, 10,068–10,074, https://doi.org/10.1029/2018GL078256, 2018.

**3    Other Changes**

A first clarification has been added in section "Preliminaries":

Adding to paraphrased text book knowledge it is here important to note that $\boldsymbol{E}^*$ is a frame-independent electrostatic field driving currents according to Equation 1. The frame-dependent motional $\boldsymbol{u} \times \boldsymbol{B}$ does not drive any currents, it is not a real field.

Changes were made according to comments by referee 2, please confirm the reply for a list.

According to my own comment in section "Preliminaries"

.

was deleted. Fukushima's contribution is reformulated as:

Fukushima (1979) had suggested that there are electric potential differences between conjugate points of only a few Volts.

References that were added are:

Cosgrove, R. B., Bahcivan, H., Chen, S., Strangeway, R. J., Ortega, J., Alhassan, M., Xu, Y., Welie, M. V., Rehberger, J., Musielak, S., and Cahill, N.: Empirical model of Poynting flux derived from FAST data and a cusp signature, Journal of Geophysical Research: Space Physics, 119, 411–430, https://doi.org/10.1002/2013JA019105, 2014.

Drob, D. P., e. a.: An update to the Horizontal Wind Model (HWM): The quiet time thermosphere, Earth and Space Science, 2, 301–319, https://doi.org/10.1002/2014EA000089, 2015.

Richmond, A. D.: On the ionospheric application of Poynting's theorem, Journal of Geophysical Research: Space Physics, 115, https://doi.org/10.1029/2010JA015768, 2010.

---

## Referee Comment (RC3) · David Knudsen (Referee) · 6 Mar 2020

NB this review is based on a version of the manuscript download in October 2019.

This paper presents and solves a simple model of magnetically conjugate neutral wind dynamos at mid latitudes. It clarifies and corrects previous descriptions and will serve as a valuable reference in the field, and I recommend that it be published. I have three substantive concerns to be addressed, and numerous suggestions for improving grammar and readability.

1) The paper discusses the motivation for using the term "entanglement" in analogy with its use in quantum mechanics. To my knowledge this term is used exclusively for a quantum mechanical effect that does not apply here. The term "coupled" is used in

circuit applications which are direct analogs of the simple system considered here, and I suggest is the more appropriate (and clearer) term.

2) In the discussion about open field lines (p14) the author states "it is doubtful that the neutral gas can act as a dynamo for the collisionless plasma in space over larger areas." As the author notes elsewhere in the paper, changes in electric field perpendicular B propagate along B as an Alfven wave. This change in electric field will change plasma drift velocity along B, will have associated with it electric currents and magnetic perturbations, and the energy content of the flux tube will change accordingly.

Consider a scenario with steady southward IMF leading to a large polar cap with open field lines. Furthermore let the solar wind speed be small so that solar-wind-driven Poynting flux into the polar cap is negligible. Next let the neutral wind in the ionosphere increase starting from zero. The result will be an Alfven wave launched upward along B, which will increase the energy density of the flux tube relative to the initial, undisturbed state. The rate of energy transfer will be associated with an upward Poynting vector, and the correct interpretation is that the neutral wind is acting as a dynamo to drive plasma motions in the collisionless region above the ionosphere. In this case the collisionless flux tube acts as a load with characteristic impedance u_0*V_A (as opposed to 1/Sigma_P in the case of a conjugate ionosphere).

I agree that it may be challenging to determine the appropriate frame in which to carry out this analysis, however I believe it is incorrect to say that the neutral wind cannot act as a dynamo on open field lines, regardless of the size of the region. I suggest that the claim quoted at the beginning of this point be removed, that the related text be removed or corrected, and that clarification of this point be left to a future communication.

3) P3, L10: it should be stated explicitly here and perhaps elsewhere that u(z) is assumed to be constant within each ionosphere. This is not clear as written.

Grammar and language usage:

P1 L11: evenly matched -> comparable (evenly matched implies they are directly competing with/opposing one another) L15: scholarly in -> in scholarly

P2: L5: with also further -> also with further (or drop "also") L7: "within two latitude circles" -> "within two constant-latitude rings" L9: in the southern hemisphere a westward (easterly) wind -> with a westward (easterly) wind in the southern hemisphere. L10: and a magnetic field aligned cartesion -> and a magnetic field-aligned cartesian L10: A ionosphere -> An ionosphere L12: interfer -> interfere L13: do play any role -> play any role L27: scholarly treated -> treated in a scholarly manner L29-31 word order: In the frame of the neutral gas in the dynamo region, roughly at altitudes of 90-350 km where collisions are significant, an electric field E* drives Pedersen and Hall currents. . ..

Figure 1 caption: allows to -> allows one to

P3: L6: top ionosphere -> topside ionosphere L7: v the ion or electron drift -> and v is the ion or electron drift L9 suggest: "For constant B, E(z) is also constant (where z is the coordinate along B). L12: request -> require L14: analogous -> analogously

P4: L2: top -> topside L11: In both, -> In both ( remove comma) L15: Galilei -> Galilean (search and replace throughout)

P5: L3: wind twice -> wind is twice

P6: L3: suggest: The title of this section, "Symmetric Dynamos", does not necessarily refer to symmetrically opposing zonal winds in an Earth-fixed frame as drawn in Figure 1 (IS THIS WHAT IS INTENDED?) L12: . . .instead of guessing them. Assumptions include:

P7: L1: The current loop between N and S closes exactly (add s to "close")

P8: A similar analysis was later performed with the Oerstedt. . . (add "the") Arguing with -> Arguing on the basis of already Fukushima (1979) -> Fukushima (1979) already. . .

P9: L7: suggest: . . .would be the result if the condition E + uxB = 0 determined E

exclusively L10+: A wind without any variations along B would not force the plasma to establish an E*, and consequently could not drive currents nor a dynamo due to zero electric field in the neutral frame.

P10: L9: but here it is an outlook for the future -> but here is left for future work. L12: convenien -> convenient L14+: Sentence beginning with "But probably more…": But more decisive factors are probably the tilt of the geomagnetic field's dipole axis, its offset from the Earth's centre, and deviations of the symmetric field with respect to the dipole equator. (Is that what is meant?)

L16: Suggest: These also cause differences near equinoxes…

L33: may only little resemble -> may only slightly resemble

P11: L4: The longitudinal dependence is indeed seen in the FAC pattern; please confirm… (use a semicolon since it separates independent clauses) L5: make it difficult -> makes it difficult L7, move "particularly" to before "consistent" ("particularly consistent"…)

L14: Ampere -> Amperes L16: "to quite consistently between" -> "quite consistently to between"

P12: L12: and does particularly not take -> and in particular does not take L15-16: with a more quantitative investigation left to a future investigation. L24: On each magnetic flux tube the neutral winds at each conjugate end provide a physical basis on which to define independent reference frames. L29: adding an in the field -> adding another definition in the field…

P13: L4: tiny delay -> small delay L11: shallow -> narrow

P14: L1: (= without collisions) -> (meaning without collisions) L6: Desired is really -> The desired expression is rather: L24: "and a neutral wind that is not constant along the magnetic field" -> "and a neutral wind that is constant within the ionosphere but different in each hemisphere.

P15: L8-9: In addition the Sq variations also reflect of course the dynamics of. . . L28: implicitely -> implicitly L32: such that explicit potential drops. . .

P16: L3: groundbased -> ground-based L13: The here presented dual entangled model -> The dual entangled model presented here L15: not restricted to dual -> not restricted to dual systems (is this what's intended?)

---

## Author Comment (AC5) · 7 Apr 2020

**Reply to David Knudsen's comment on "Entangled Dynamos and Joule Heating in the Earth's Ionosphere" by**

Stephan C. Buchert[1]

[1]Swedish Institute of Space Physics, Uppsala, Sweden

**Correspondence:** Stephan Buchert (scb@irfu.se)

**1   Replies to comments by Referee 3**

Cited referee comments are in red, replies in magenta.

I thank the referee, David Knudsen, for the interest in the manuscript and the time spent reading it, and for the helpful comments.

1) The paper discusses the motivation for using the term "entanglement" in analogy with its use in quantum mechanics. To my knowledge this term is used exclusively for a quantum mechanical effect that does not apply here. The term "coupled" is used in circuit applications which are direct analogs of the simple system considered here, and I suggest is the more appropriate (and clearer) term.

**Reply:**

"Coupled" in "coupled dynamos" would not be a good adjective. An essential point is that the dynamos only exist because of the mismatched or not mapping neutral winds at conjugate points in a dipole-like magnetic field configuration. "Uncoupled" there are no dynamos. This is different from a ionosphere and a magnetosphere which may exist independently (examples are Venus and Mars having only ionospheres and Mercury having only a magnetosphere), but can be coupled for planets like the Earth. Therefore I don't want a title having "coupled dynamos". A title like "Interhemisphere coupling of the ionosphere-thermosphere and Joule heating" seems too general, unspecific.

There are cases in classical physics where "entangled" is being used, for example Islam and Archer (2001), "Non-linear rheology of highly entangled polymer solutions in start-up and steady shear flow.".

The original word used by Schrödinger is the German "verschränkt", which can alternatively be translated to English as "crossed."

An alternative translation is "crossed", like in "crossed arms". "Crossed" would also be an approriate word describing here "crossed dynamos". There are similarities with entangled states known from quantum mechanics:

has been added in section 5.1

It is true that essential features of quantum mechanical entanglement do not apply here: There is no quantization and no probability interpretation. For example, by considering another spatial direction John Bell arrived at his

famous inequality predicting the statistical outcome of a large number of measurements. Here one will have to consider not only zonal, but also meridional winds to establish a more realistic model. This is mentioned in the manuscript. But the outcome of taking into account the other spatial direction will certainly not resemble Bell's inequality in any way, as probabilities aren't involved. However, as mentioned in the manuscript, the "entangled dynamos" do have an element of action at a distance which, in the quantum mechanical case, Einstein had called "spooky" (="gespenstisch").

2) In the discussion about open field lines (p14) the author states "it is doubtful that the neutral gas can act as a dynamo for the collisionless plasma in space over larger areas." As the author notes elsewhere in the paper, changes in electric field perpendic- ular B propagate along B as an Alfven wave. This change in electric field will change plasma drift velocity along B, will have associated with it electric currents and magnetic perturbations, and the energy content of the flux tube will change accordingly.

Consider a scenario with steady southward IMF leading to a large polar cap with open field lines. Furthermore let the solar wind speed be small so that solar-wind-driven Poynting flux into the polar cap is negligible. Next let the neutral wind in the ionosphere increase starting from zero. The result will be an Alfven wave launched upward along B, which will increase the energy density of the flux tube relative to the initial, undis- turbed state. The rate of energy transfer will be associated with an upward Poynting vector, and the correct interpretation is that the neutral wind is acting as a dynamo to drive plasma motions in the collisionless region above the ionosphere. In this case the collisionless flux tube acts as a load with characteristic impedance $\mu_0 * V_A$ (as opposed to $1/Sigma_P$ in the case of a conjugate ionosphere).

I agree that it may be challenging to determine the appropriate frame in which to carry out this analysis, however I believe it is incorrect to say that the neutral wind cannot act as a dynamo on open field lines, regardless of the size of the region. I suggest that the claim quoted at the beginning of this point be removed, that the related text be removed or corrected, and that clarification of this point be left to a future communication.

**Reply:** The statement that the the neutral wind cannot act as a dynamo on open field lines refers to a steady state. The manuscript generally describes only the steady state as mentioned in the introduction. The statement has been changed to

. . . doubtful that the neutral gas can act as dynamo for the collisionless plasma in space in a steady state over larger areas. Temporal variations of a neutral wind would in principle excite Alfvén waves adjusting the mechanical stress balance between ionosphere-thermosphere and space plasma which, however, does not lead to any dynamo driven dissipation in space.

The situation described by the referee is not a steady state, and is certainly not the explanation for the average upward Poynting flux found in satellite data. The ionosphere-magnetosphere system on open field-lines readjusts very quickly (by transmitting an Alfvén waves) and reaches a new quasi-steady state. Unless there would a continuous sufficiently rapid temporal change of the neutral wind which the large inertia of the neutral gas prevents.

On open field-lines steady state current systems are involved in an exchange of momentum between Earth and the magnetosphere, see also Vasyliūnas (2007). This is a mechanical process based on Newton's second law, the conservation of momentum (not energy). Currents and the $\boldsymbol{j} \times \boldsymbol{B}$ force are independent of the reference frame (in the non-relativistic limit). Undisputably this process takes place in a quasi-steady state. A consistent pattern of the FAC shows up when averaging a large amount of satellite measurements (Iijima and Potemra, 1976).

In some publications it is stated, that this process of mechanical momentum transfer changes the kinetic energy in the ionosphere-thermosphere. This consideration, however, is frame dependent. Any chosen frame would arbitrarily define how much kinetic energy is in the ionosphere-thermosphere, and whether the momentum exchange between ionosphere and magnetosphere increases or decreases it. Therefore, after realising the inherent frame dependence of the $\boldsymbol{u} \times \boldsymbol{B}$ field, I have stayed away in the manuscript from a discussion of the kinetic energy.

Relevant is rather the conversion to thermal energy. This energy is frame independent, and the conversion, in the thermodynamic sense, is irreversible. The space plasma is generally assumed to be collisionless. Still dissipation, i.e. conversion to thermal energy, can take place at special locations. A prominent example is the bow shock. However, it is not plausible that the neutral atmosphere is in any way connected to such processes.

Thus, returning to the scenario of the referee, the thermal energy density of a flux tube with collisionless plasma does not increase because of a neutral wind at the bottom. This is consistent with the electric field and the Poynting flux in the frame of the plasma being zero. If there is a temporal change, as noted by the referee, then Alfvén waves are generated. After the wave has faded away and a new steady state is reached the thermal energy density of the plasma on the flux tube would be unchanged, and there has been no dynamo action by the neutral wind. But an exchange of momentum between ionosphere-thermosphere and space plasma has taken place.

3) P3, L10: it should be stated explicitly here and perhaps elsewhere that u(z) is assumed to be constant within each ionosphere. This is not clear as written.

**Reply:**

$\boldsymbol{u}$ and $\boldsymbol{B}$ are also assumed constant over the altitude range where there is significant collisional interaction with the plasma. In other words, the ionosphere is assumed to be thin.

has been added.

**Grammar and language usage:**

P1 L11: evenly matched -> comparable (evenly matched implies they are directly competing with/opposing one another) L15: scholarly in -> in scholarly  changed as suggested.

P2: L5: with also further -> also with further (or drop "also") L7: "within two latitude circles" -> "within two constant-latitude rings" L9: in the southern hemisphere a westward (easterly) wind -> with a westward (easterly) wind in the southern hemisphere. L10: and a magnetic field aligned cartesion -> and a magnetic field-aligned cartesian L10: A ionosphere -> An ionosphere L12: interfer -> interfere L13: do play any role -> play any role L27: scholarly treated -> treated in a scholarly manner L29-31 word order: In the frame of the neutral gas in the dynamo region, roughly at altitudes of 90-350 km where collisions are significant, an electric field E* drives Pedersen and

Hall currents. ... changed as suggested, except for using "circles of latitude" instead of "constant-latitude rings" because it is a fixed expression in geodesy (https://en.wikipedia.org/wiki/Circle_of_latitude)

Figure 1 caption: allows to -> allows one to changed as suggested.

P3: L6: top ionosphere -> topside ionosphere L7: v the ion or electron drift -> and v is the ion or electron drift L9 suggest: "For constant B, E(z) is also constant (where z is the coordinate along B). L12: request -> require L14: analogous -> analogously changed as suggested.

P4: L2: top -> topside L11: In both, -> In both ( remove comma) L15: Galilei -> Galilean (search and replace throughout) changed as suggested.

P5: L3: wind twice -> wind is twice changed as suggested.

P6: L3: suggest: The title of this section, "Symmetric Dynamos", does not necessarily refer to symmetrically opposing zonal winds in an Earth-fixed frame as drawn in Figure 1 (IS THIS WHAT IS INTENDED?) changed to:

The title of this section "Symmetric Dynamos" does not refer to the zonal winds that are symmetrically opposing in an Earth fixed frame as drawn in Figure 1. The same results are obtained for any wind difference that is equal to this symmetric case. "Symmetric" rather refers to ...

The point here is the insight that the absolute winds, symmetric in a certain reference frame or not, are irrelevant. Only the wind difference is important.

L12: ...instead of guessing them. Assumptions include:

Requirements that apply for both the symmetric and asymmetric cases include:

has been added.

P7: L1: The current loop between N and S closes exactly (add s to "close") changed as suggested.

P8: A similar analysis was later performed with the Oerstedt. . . (add "the") Arguing with -> Arguing on the basis of already Fukushima (1979) -> Fukushima (1979) already ... changed as suggested.

P9: L7: suggest: . . .would be the result if the condition E + uxB = 0 determined E exclusively L10+: A wind without any variations along B would not force the plasma to establish an E*, and consequently could not drive currents nor a dynamo due to zero electric field in the neutral frame. changed as suggested.

P10: L9: but here it is an outlook for the future -> but here is left for future work. L12: convenien -> convenient L14+: Sentence beginning with "But probably more. . .": But more decisive factors are probably the tilt of the geomagnetic field's dipole axis, its offset from the Earth's centre, and deviations of the symmetric field with respect to the dipole equator. (Is that what is meant?) changed as suggested (by referee 1).

L16: Suggest: These also cause differences near equinoxes ... changed as suggested.

L33: may only little resemble -> may only slightly resemble changed as suggested.

P11: L4: The longitudinal dependence is indeed seen in the FAC pattern; please confirm . . . (use a semicolon since it separates independent clauses) L5: make it difficult -> makes it difficult L7, move "particularly" to before "consistent" ("particularly consistent" . . . ) changed as suggested.

L14: Ampere -> Amperes L16: "to quite consistently between" -> "quite consistently to between" changed as suggested.

P12: L12: and does particularly not take -> and in particular does not take L15-16: with a more quantitative investigation left to a future investigation. L24: On each magnetic flux tube the neutral winds at each conjugate end provide a physical basis on which to define independent reference frames. L29: adding an in the field -> adding another definition in the field . . . changed as suggested.

P13: L4: tiny delay -> small delay changed as suggested.

L11: shallow -> narrow "shallow" means here changing slowly, with a very small derivative/slope, because the distance along the field-line through the plasmasphere over which the change occurs is very large. So I did not change "shallow".

P14: L1: (= without collisions) -> (meaning without collisions) L6: Desired is really -> The desired expression is rather: L24: "and a neutral wind that is not constant along the magnetic field" -> "and a neutral wind that is constant within the ionosphere but different in each hemisphere. changed as suggested.

P15: L8-9: In addition the Sq variations also reflect of course the dynamics of . . . L28: implicitely -> implicitly L32: such that explicit potential drops . . . changed as suggested, or text deleted following a comment by referee 2.

P16: L3: groundbased -> ground-based L13: The here presented dual entangled model -> The dual entangled model presented here L15: not restricted to dual -> not restricted to dual systems (is this what's intended?)

The text is now:

A three-way entanglement of the dynamos in the equatorial F and E regions might turn out to be an applicable concept.

**2 References**

Iijima, T., and Potemra, T. A. ( 1976), The amplitude distribution of field-aligned currents at northern high latitudes observed by Triad, J. Geophys. Res., 81( 13), 2165– 2174, doi:10.1029/JA081i013p02165.

Islam, M.T. and Archer, L.A. (2001), Nonlinear rheology of highly entangled polymer solutions in start-up and steady shear flow. J. Polym. Sci. B Polym. Phys., 39: 2275-2289. doi:10.1002/polb.1201

Vasyliūnas, V. M.: The mechanical advantage of the magnetosphere: solar-wind-related forces in the magnetosphere-ionosphere-Earth system, Ann. Geophys., 25, 255–269, https://doi.org/10.5194/angeo-25-255-2007, 2007.

**3 Other Changes**

The manuscript has been changed according to referees 1 and 2 comments and my replies to these comments.

---

## Author Response (AR2)

**Reply to the comment by Referee 1's on "Entangled Dynamos and Joule Heating in the Earth's Ionosphere" by**

Stephan C. Buchert[1]

[1]Swedish Institute of Space Physics, Uppsala, Sweden

**Correspondence:** Stephan Buchert (scb@irfu.se)

The reference Park and Lühr (2020) should read Park, Yamazaki and Lühr (2020), see https://doi.org/10.1029/2019JA027694, 2020.

Reply: The author list in the *bib file was incorrectly formatted, and so BibTeX interpreted "Yamazaki" as a given name of Park. Thanks for spotting this.

**Reply to the comment by Referee 2's on "Entangled Dynamos and Joule Heating in the Earth's Ionosphere" by**

Stephan C. Buchert[1]

[1]Swedish Institute of Space Physics, Uppsala, Sweden

**Correspondence:** Stephan Buchert (scb@irfu.se)

a1) p. 7, Eq. (5): The minus sign should be dropped from the expression of $J_S$. Thus, both $J_S$ and $E_S^*$ will have the same orientation, consistent with Figure 4.

a2) p. 7, Eq. (6) and L21: The *absolute value* of the Pedersen current is the same in both hemispheres (slight adjustment needed at L21), though the sign is different, because of the opposite orientations of the respective X axes. Thus, $J_N = -J_S => J_N + J_S = \ldots = 0$.

Reply: Yes, since $\boldsymbol{E}_{N,S}^*$ and $\boldsymbol{J}_{N,S}$ are vectors, I should have been more careful with the signs.

$\boldsymbol{E}_N^*$ and $\boldsymbol{E}_S^*$ have opposite north-south directions (for purely zonal winds), because of the orientation of the X axis in each hemisphere, and the field components $E_N^*$ and $E_S^*$ have opposite signs:

has been added. Eq. (5) is now

$$J_N = \Sigma_N E_N^*, \ J_S = \Sigma_S E_S^*;$$

$$J_N + J_S = \Sigma_N E_N^* + \Sigma_S E_S^* = 0$$

b) p.2, L33: I think collisions and ionospheric currents are most significant in the lower part of the 90-350 km altitude range, below some 150-200 km.

Reply: I changed 90–350 km to 90–200 km. However, especially at equatorial latitudes there is a signficant "F-region dynamo". Also this F-region dynamo cannot be driven by neutral winds in the F region per se, but rather must be caused by wind differences. This is briefly mentioned in the last sentence of the manuscript. For a future investigation of especially this subject dynamo effects up to at least 350 km altitude need to be taken into account.

c) p. 13, L5: the describe => to describe; confirm Figure 1 => conform Figure 1 (?)

Reply: Thanks for spotting. The text is changed to:

[revised manuscript text omitted]